# Thermal effect on the fecundity and longevity of *Bactrocera dorsalis* adults and their improved oviposition model

**Kyung San Choi**[1], **Ana Clariza Samayoa**[2], **Shaw-Yhi Hwang**[2], **Yu-Bing Huang**[3], **Jeong Joon Ahn**[1]*

**1** Research Institute of Climate Change and Agriculture, National Institute of Horticultural and Herbal Science, Rural Development Administration, Jeju, Republic of Korea, **2** Department of Entomology, National Chung Hsing University, Taichung, Taiwan (R.O.C), **3** Taiwan Agricultural Research Institute, Taichung, Taiwan (R.O.C)

* j2ahn33@korea.kr

**Data Availability Statement:** All relevant data are within the manuscript and its Supporting Information files.

## Abstract

The oriental fruit fly, *Bactrocera dorsalis*, is a destructive polyphagous pest that causes damage to various fruit crops, and their distribution is currently expanding worldwide. Temperature is an important abiotic factor that influences insect population dynamics and distribution by affecting their survival, development, and reproduction. We examined the fecundity, pre-oviposition and oviposition periods, and longevity of adult *B. dorsalis* at various constant temperatures ranging from13°C to 35°C. The longevity of female *B. dorsalis* ranged from 116.8 days (18.8°C) to 22.4 days (34.9°C), and the maximum fecundity per female was 1,684 eggs at 28.1°C. Females were only able to lay eggs at 16.7°C to 34.9°C, and both the pre-oviposition and oviposition periods were different depending on the temperature. We modeled female reproduction in two oviposition models (OMs): 1) the current model developed by Kim and Lee, an OM composed of a fecundity model, age-specific survival model, and age-specific cumulative oviposition rate model, and 2) a two-phase OM modified the logic structure of the current model by separating pre-oviposition, so that oviposition was estimated with the female in oviposition phase who had complete pre-oviposition phase. The results of the two-phase OM provided more realistic outputs at lower and higher temperatures than those of the current model. We discussed the usefulness of the two-phase OM for the reproduction of insects with long pre-oviposition periods.

## Introduction

The oriental fruit fly *Bactrocera dorsalis* (Hendel) (Diptera: Tephritidae) is a destructive polyphagous pest that damages various fruits by piercing the skin for oviposition and consequent fruit destruction caused by hatched larva. Therefore, *B. dorsalis* also cause economic damage to the food industry due to market loss or reduction [1–2]. Their distribution has rapidly expanded from their location of origin in South Asia to northern and eastern nations and islands, including Taiwan, Japan, and the Pacific Islands in the 1900s [2, 3–6]. In Taiwan,

**Funding:** Dr Shaw-Yhi Hwang was received funds from Rural Development Administration and the study was carried out with the cooperative research project (Project number: PJ01207501) between National Chung Hsing University (NCHU), Taiwan and National Institute of Horticultural and Herbal Science (NIHHS), Rural Development Administration (RDA), Republic of Korea.

**Competing interests:** The authors have declared that no competing interests exist.

*B. dorsalis* became a major pest with periodic occurrences throughout the state since its initial discovery in 1911 [7–8]. Area-wide management by annihilating the males of the species with effective methyl-eugenol attractants and insecticides reduced fruit damage and loss in several crops [9–10]. Japan succeeded in eradicating *B. dorsalis* in the 1980s [11], but subsequently faced persistent invasion from neighboring nations [12]. In China, although *B. dorsalis* has always been a dominant species [13], its distribution has continually expanded to mid-China below the overwintering limitation altitude of 30˚N [14–16]. Although *B. dorsalis* adults have long-distance flight ability, enabling them to forage as far as 24 miles away [17], their settlement is greatly limited by the temperature and available host plants. The oriental fruit fly feeds on approximately 300 plant species including crop and wild plants [18]. *B. dorsalis* normally develop and survive in a temperature range of 15 ˚C to 35˚C, unless they experience mass mortality and are unable to develop, inferred from the studies on relationship between temperature and development and survival [14, 19–26]. Several studies have investigated the reproduction of *B. dorsalis* in relation to altering temperatures, food resources, dietary restrictions, polyandry, and glutamine synthetase [1, 27–32]. However, there have been fewer studies on modeling adult reproduction (i.e., oviposition model) compared to modeling the immature stage's development. Several approaches exist for constructing oviposition models (OMs) for many arthropod pests based on the logical structure developed by Kim and Lee [33] such as *Carposina sasakii*, *Tetranychus urticae*, *Otiorhynchus sulcatus*, *Scotinophara lurida*, *Riptortus pedestris*, *Plutella xylostella*, *Neoseiulus californicus*, *Cnaphalocrocis medinalis*, *Trissolcus basalis*, *Scirtothrips dorsalis*, *Ephestia kuehniella* and *Rhopalosiphum padi* [33–45]. The OM comprises three essential temperature-dependent components: total fecundity, age-specific oviposition rate and age-specific survival rate. Total fecundity model uses temperature as an input and these two age-specific models use physiological age as an input that is a sum of the outputs from female aging model with mean temperatures of each day from adult emergence. Therefore, survival and oviposition rate at a day depend on the daily temperatures that the female cohort had experienced after their emergence. According to the fecundity model, total capacity of oviposition is determined by the temperature condition when the female starts oviposition. The OM have a compact logic structure to simply predict egg occurrence of insect species that has a short life span as *C. sasakii*. The OM of *C. sasakii* was incorporated into the population model of an orchard system for establishing management strategies [46].

In this study, we investigated the effect of temperature on the fecundity and longevity of *B. dorsalis* adults. *B. dorsalis* has a distinctive pre-oviposition period and longevity of *B. dorsalis* adult is much longer than that of *C. sasakii* depending on temperature. Therefore, we developed a modified model that can be applied to an insect with long life span as well as to precisely describe the relationship between temperature and oviposition (Fig 1). The modified model will contribute towards predictions of the seasonal occurrence and oviposition of *B. dorsalis*.

## Materials and methods

### Insect colony

*Bactrocera dorsalis* pupae were provided by the Taiwan Agricultural Research Institute (TARI). The populations were originally collected from a wild orchard in Wufeng county, Taichung, Taiwan. They were reared on an artificial diet for more than 200 generations in TARI. Insects were reared according to the method of Huang and Chi [30] based on an artificial diet developed by Tanaka et al. [47]. Pupae were placed in a plastic netted cage (30 x 30 cm) and emerged as adults, who were then fed an artificial diet composed of 200 g Yeast Hydrolysate Enzymatic (MP Biomedicals, LLC., Illkirch-Graffenstaden, France), 40 g granulated sugar, 10

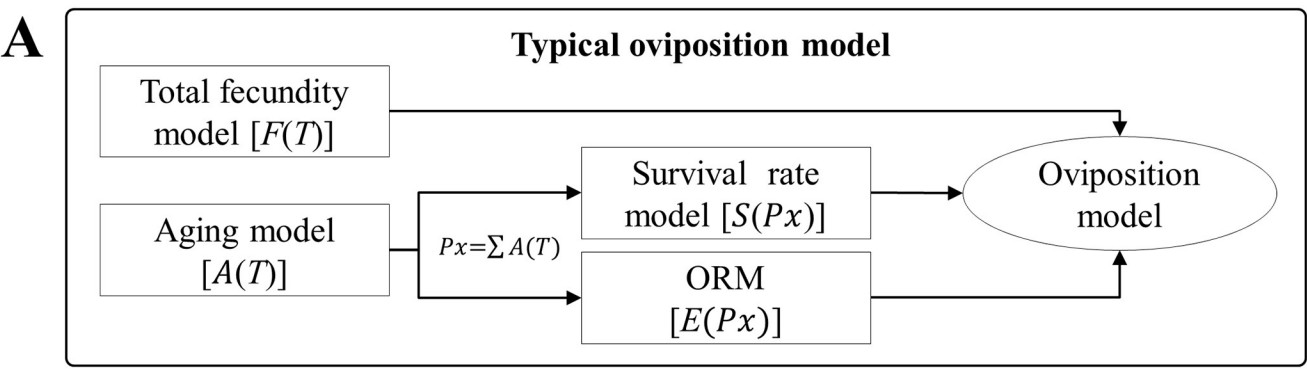

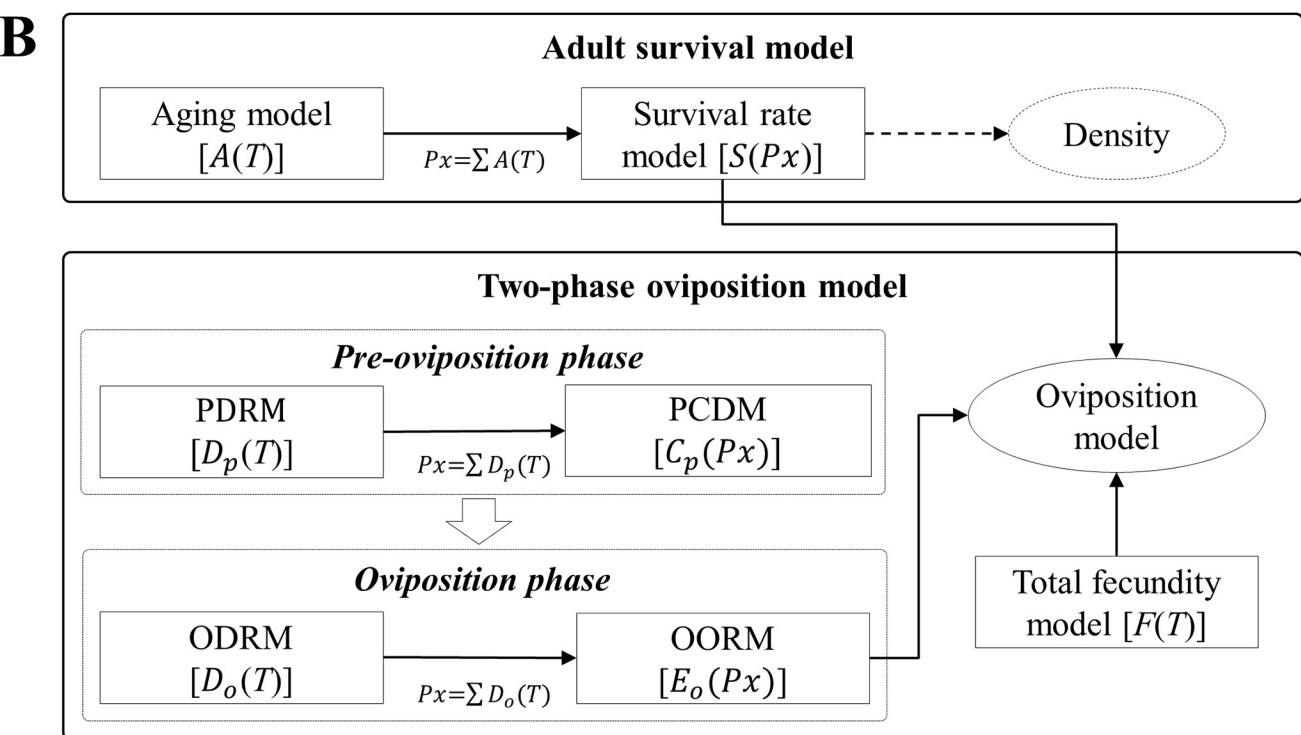

**Fig 1. Illustration of the model structure and simulation process for the current oviposition model (OM) (A) and two-phase OM (B).** (A): Current OM is composed of three components; fecundity model and two age-specific models, female survival rate model and cumulative oviposition rate model (ORM) based on physiological age (*Px*) derived from the female aging model. (B): Two-phase OM has two separate phases: The pre-oviposition phase is for newly emerged females and a portion of the females that completed the pre-oviposition phase is predicted by a pre-oviposition complete distribution model (PCDM) based on the physiological age derived from a pre-oviposition development rate model (PDRM). The oviposition phase predicts the daily oviposition of the female who completed pre-oviposition with an age-specific cumulative oviposition rate model of the oviposition phase (OORM) based on the physiological age derived from the oviposition development rate model (ODRM). Finally, the daily egg production laid by the females after emergence was predicted by the product of the daily proportion of egg production, fecundity model, and survival model.

ml protein hydrolysate (Alco Standard Co., Pennsylvania, USA), and 50 ml water. For the oviposition site, a plastic cylinder (4.5 cm diameter and 5 cm height) containing a 20% guava juice cotton ball was placed into the cage. Eggs were laid inside the cylinder and were collected using distilled water. Subsequently, approximately 4 mL of eggs were inoculated into a container (90 x 15 mm) filled with an artificial diet composed of 5 g sodium benzoate (Sigma-Aldrich Co. Milan, Italy), 240 g granulated sugar (Taiwan Sugar corporation, Tainan, Taiwan),

140 g yeast (Vietnam—Taiwan Sugar Co., Thanh Hoa, Vietnam), 20 mL HCI (Sigma-Aldrich Co., Austria), 480 g wheat grain (purchased at Taichung), and 1100 ml of distilled water. Matured larvae left the container and pupated in the surrounding sawdust. Pupae were collected by sieving the sawdust.

### Laboratory experiment

The fecundity and longevity of *B. dorsalis* adults were examined in growth chambers (Model A 414931206, Yuh Chuen Chiou Industry Co., Kaohsiung, Taiwan) set to seven constant temperatures; 13˚C, 16˚C, 20˚C, 24˚C, 28˚C, 32˚C, and 35˚C, with a 14: 10 h (light: dark) photoperiod. Temperature and humidity in the chamber were recorded at 1 h intervals by a data logger (HOBO, ONSET computer, Co., USA). The humidity range was 50~70% at all temperatures except the chamber set to 20˚C, where it was 19~30%.

One newly emerged (<12 h) virgin female and two males were placed into a cage ($10 \times 15 \times 10$ cm) with a supply of the artificial diet and a 10% sugar- containing gel. Eighteen or twenty cages were treated at each temperature, but a few cages were excluded in analysis when female escaped from a cage during egg examination. A female with two males produces significantly more eggs than females with only one male [30]. A perforated plastic cup (diameter and height both 4 cm) was provided for the oviposition site with a cotton ball soaked in guava juice placed inside the cup. The number of eggs laid per female in the cup was recorded daily. The adult oviposition period (AOP) of each female was determined by the day from which the female first laid a significant number (>5) of eggs consistently, as many females did not lay the eggs consistently at 13.5˚C, 16.7˚C and 34.9˚C. Adult pre-oviposition period (APOP) was defined from adult emergence to AOP, and total pre-oviposition period (TPOP) was obtained by summing the APOP and immature development period examined in previous study [26].

An analysis of variance was conducted to determine the statistical differences in adult longevity, fecundity, APOP, AOP, and TPOP using SAS [48] after checking normality of the data by Skewness, Kurtosis, and Kolmogrov Smirov methods. Means were separated using Tukey's honest significance test (HSD; P = 0.05).

### Low developmental threshold and thermal constant

Aging rates (1/mean days of longevity) of both males and females were plotted against temperature, respectively (Fig 2A and 2B, S1 and S2 Tables). A linear regression was conducted in the region where rates increased linearly (over 18.8˚C). The slope (a) and intercept (b) of the linear model ($Y = aX + b$) were estimated using the Table Curve 2D program [49], and were used to deduce the low development threshold (LDT) and thermal constant (TC) by solving–b/a and 1/a, respectively. LTD is the temperature below which development stops. TC provides a measure of the physiological time required for the completion of a development process and is measured in degree-day (DD) and is a product of time and the degrees of temperature above the threshold temperature. A linear regression test was applied to the region showing a proportional relationship between temperature and development rates of APOP, AOP and TPOP, respectively.

### Non-linear model development

**Physiological age.** The physiological age ($Px$) was obtained by accumulating the rates computed with a development model [$D(T)$] as an input of temperature $T_i$˚C from the starting day (0) to the $n$th day [50]. Development models in this study are adult aging models and

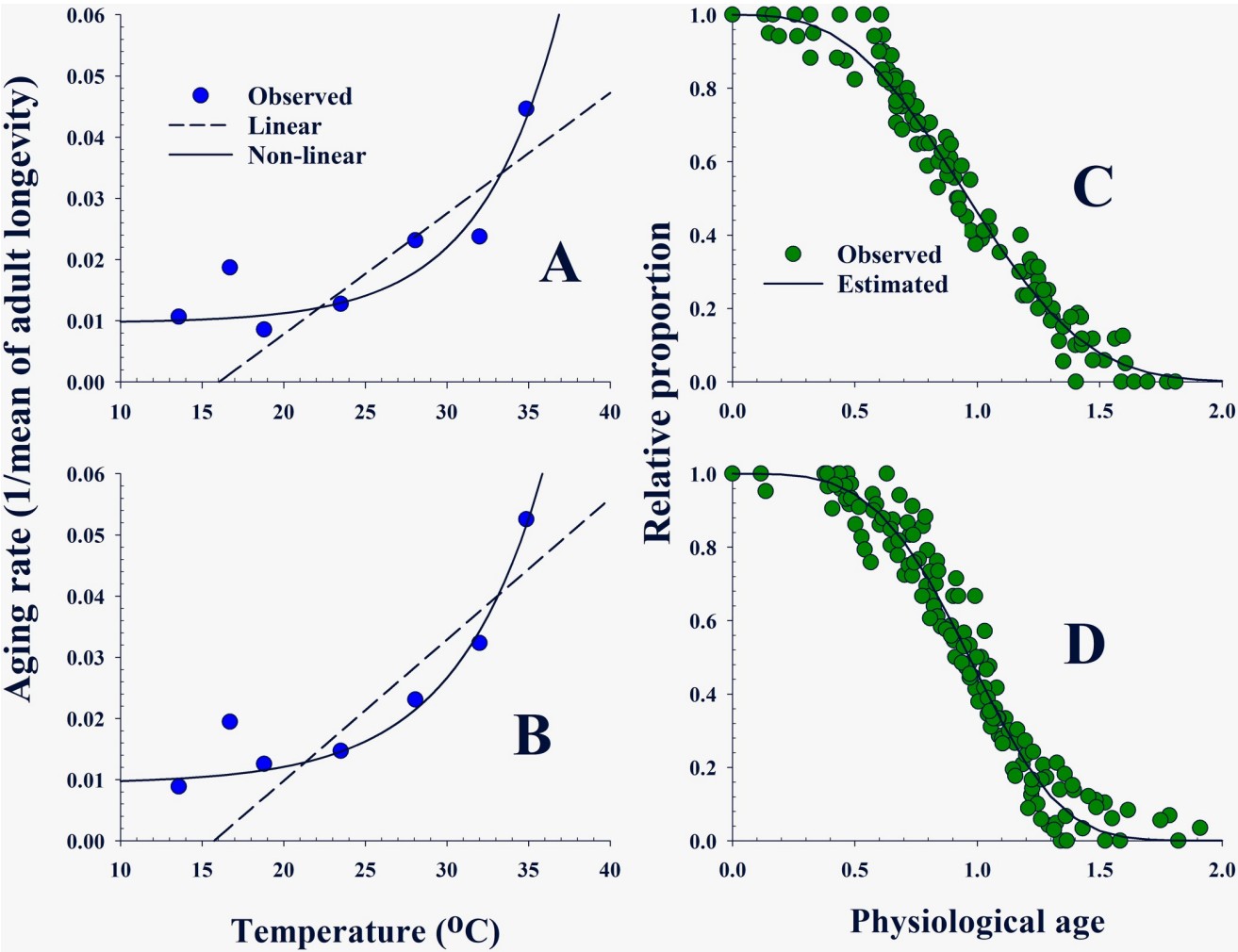

**Fig 2. Aging and survival rates of *Bactrocera dorsalis* adult females and males at various constant temperatures.** (A) and (B): A linear model (dot lined) and a non-linear model (solid line) were fitted to the aging rate (1/mean days of longevity) curve of the females and males, respectively. (C) and (D): Two-parameter Weibull function was applied to the respective survival proportion curves of females and males, respectively, based on their physiological age.

development rate models for pre-oviposition and oviposition phase, respectively (Fig 1)

$$Px = \int_{t_0}^{t_n} D[T(t)] \, dt \approx \sum_{i=0}^{n} D(T_i) \, \Delta t \tag{1}$$

**Aging model.**   The aging rates (1/mean days of longevity) of both male and female adults were stationary under 20°C but then increased exponentially. The obtained flat then increasing curve is similar to the aging rate curve previously obtained for *Ascotis selenaria* [51]. We applied the function to describe the aging rate curves of males and females, respectively:

$$A(T) = r_0 + exp\left[\frac{(T - T_H)}{c}\right] \tag{2}$$

where $A(T)$ is the aging rate at $T$ °C with a minimum aging rate ($r_0$) and $\approx 1$ at $T_H$°C.

**Age-specific survival rate model.** The age-specific survival rate presents the proportion alive at any given time. The number of surviving adult male and female at each day after adult emergence in each temperature tested was calculated as the relative survival proportion to the total number of examined adults at each temperature. Adult age was expressed in days spent at the specified temperature, and was normalized as a physiological age by dividing the number of days by the mean longevity of the temperature, which finally formed the age-specific probability distribution (Fig 2C and 2D, S3, S4, S5 and S6 Tables). A modified two-parameter Weibull equation [52] was applied to compute relative survival ($S(Px)$) at a physiological age ($Px$):

$$S(Px) = \exp[-(Px/\alpha)^{\beta}] \tag{3}$$

Parameter $\alpha$ indicates the physiological age of 50% of surviving individuals and $\beta$ is a shape parameter of the curve.

**Total fecundity model.** The relationship between total fecundity per female and temperature was calculated by the equation proposed by Briere et al [53]. Eggs were laid between 16.7°C and 34.9°C with a specifically high amount at 24°C and 28°C, shaping a campaniform (Fig 3A, S7 Table):

$$F(T) = \alpha \cdot (T - T_L) \cdot (T_H - T) \tag{4}$$

where $F(T)$ is the total number of eggs that a female is capable of laying in her life-span at a temperature $T$°C with low ($T_L$) and high ($T_H$) temperature limits. $\alpha$ is the empirical constant of the equation.

**Age-specific cumulative oviposition rate model (ORM).** The number of eggs laid by the female in a day after adult emergence in all examined temperatures was translated by normalizing with the physiological age, as described in the survival model, into the age-specific cumulative oviposition probability distribution (Fig 3B, S8 and S9 Tables). Three-parameter Weibull function [52,54] was applied:

$$E(Px) = 1 - \exp\left[-\left(\frac{Px - \gamma}{\eta}\right)^{\beta}\right] \tag{5}$$

where $E(Px)$ is the cumulative proportion of eggs laid by the female at a physiological age ($Px$). Parameter $\gamma$ is a physiological age when the first egg appeared in this model, and $\eta$, and $\beta$ are parameters of the equation.

**Pre-oviposition development rate model (PDRM) and completion distribution model (PCDM).** We divided female life into two phases, pre-oviposition and oviposition. The pre-oviposition development rate (1/mean days of pre-oviposition period) of *B. dorsalis* increased linearly as temperature increased up to 32°C, then decreased at 34.9°C (Fig 4A, S10 Table). The Briere 2 model [53] was applied to describe the relationship between temperature and development rate:

$$D_p(T) = \alpha \cdot T \cdot (T - T_L) \cdot (T_H - T)^{\frac{1}{m}} \tag{6}$$

where $D_p(T)$ is the development rate model of pre-oviposition and $m$ is the empirical constant of the equation.

The pre-oviposition period of the females at different temperatures was translated to the age-specific cumulative completion distribution as described in the survival model (Fig 4C, S11 and S12 Tables). The three-parameter Weibull function in Eq (5) was applied to PCDM [$C_p(Px)$] to predict the probability of the female completing pre-oviposition at a physiological age ($Px$).

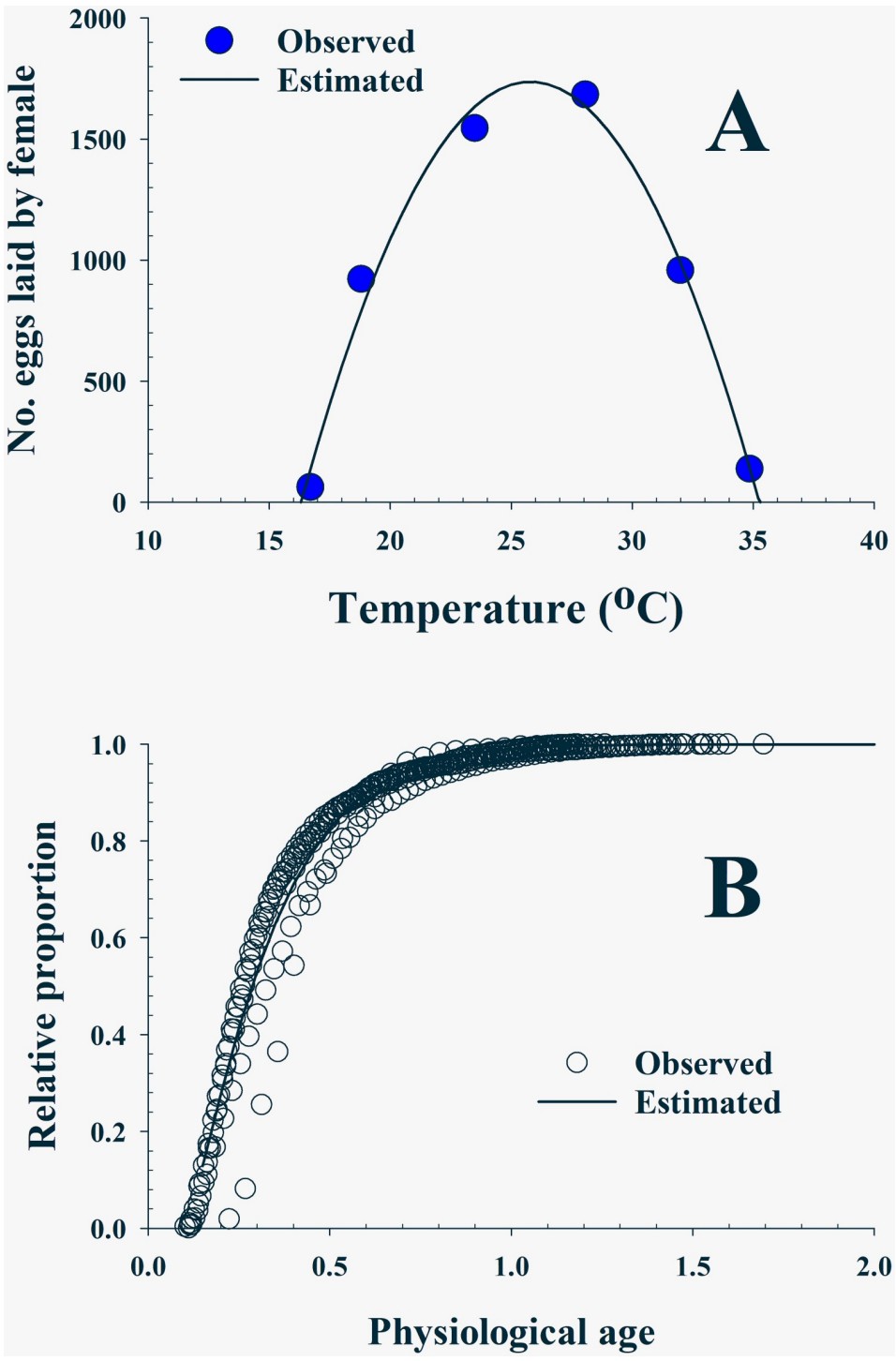

**Fig 3.** (A): Fecundity of *Bactrocera dorsalis* at various temperatures where a quadratic equation (line) was applied. (B): Three-parameter Weibull function was applied to the cumulative oviposition probability distribution of the female after emergence.

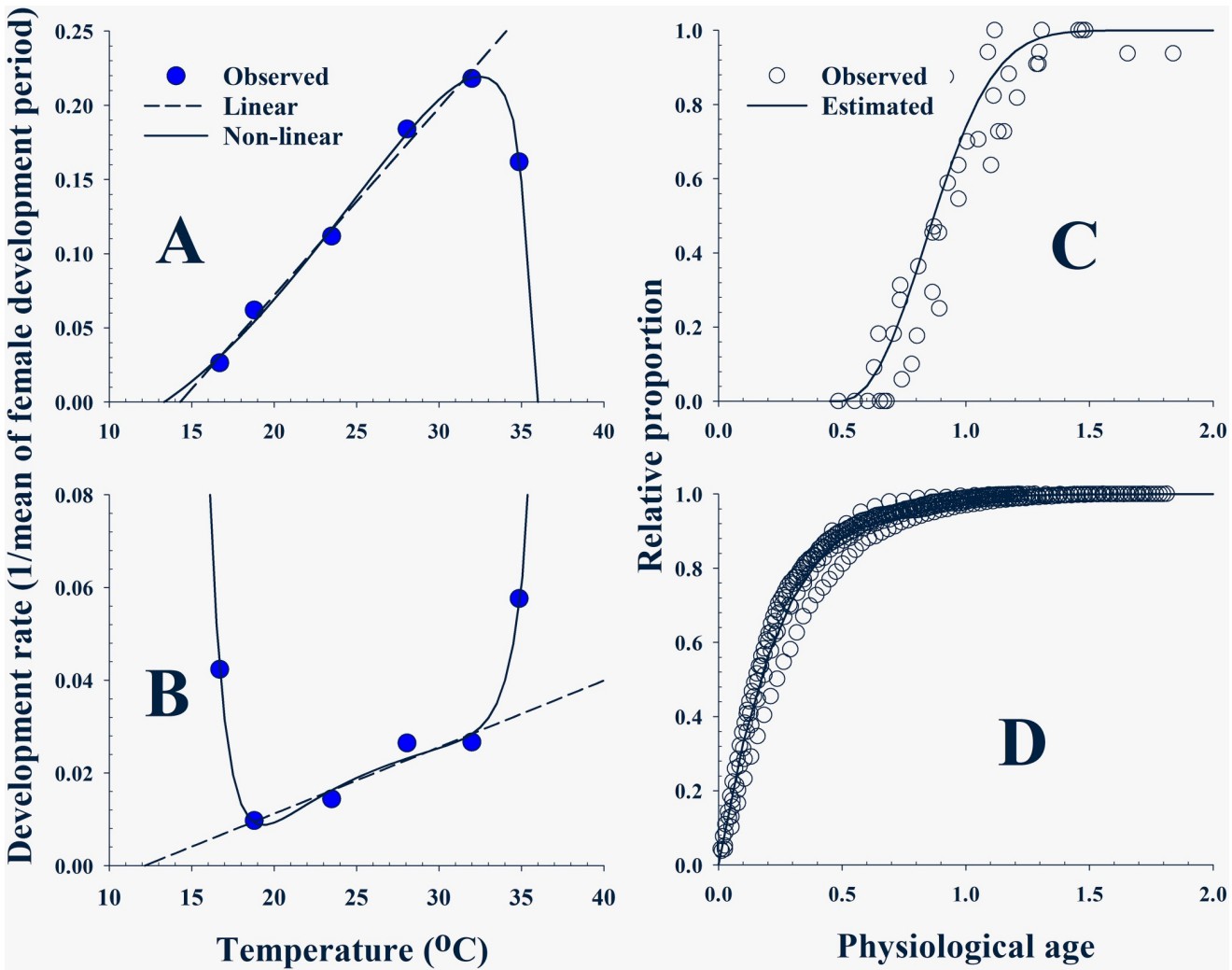

**Fig 4.** (A) and (B): Development rate (1 / mean days) curves of the *Bactrocera dorsalis* female in the pre-oviposition and oviposition periods in which the Biere 2 model and a non-linear model were applied to pre-oviposition and oviposition periods, respectively. (C) and (D): Three-parameter and two-parameter Weibull functions were applied, respectively, to describe the cumulative proportion of the females that completed pre-oviposition and their cumulative oviposition probability in oviposition phase based on their physiological age.

**Oviposition development rate model (ODRM) and age-specific cumulative oviposition rate model of oviposition phase (OORM).** The development rates (1/mean days of oviposition period) of the oviposition period showed a slowly increasing region from 16.7˚C to 32.0˚C, and steeply increased at either the end of the low or high temperatures (Fig 4B, S13 Table). A model showing a curve of best fit was selected for ODRM [$D_o(T)$] that computes a development rate of a female in oviposition phase at a temperature $T$ ˚C.

$$D_o(T) = a + b \cdot \exp(T) + c/T^2 + d \cdot \exp(-T) \tag{7}$$

Eggs laid by the female were translated to the age-specific cumulative oviposition probability distribution (Fig 4D, S14 and S15 Tables). The two-parameter Weibull function used in Eq (3) was applied to compute the cumulative proportion of eggs laid by a female at a physiological age ($Px$) in oviposition phase.

**Model selection and parameter estimation.**  The Table Curve 2D program [49] was used to select a best-fit model for ODRM, but was mostly used for parameter estimation of the models through statistical analysis when its curve was best fit to corresponding observed or translated data.

## Oviposition models and their simulation

**Oviposition model (OM).**  Kim and Lee [33] developed an OM composed of a fecundity model and two age-specific models, ORM and female survival rate model, based on the physiological age derived from the female aging model (Fig 1A). We used a modified OM where the female survival expression $[(S(Px_i) + S(Px_{i-1}))/2]$ of the OM expression was simplified by Choi and Kim [51] as:

$$F(T) \cdot [E(Px_i) - E(Px_{i-1})] \cdot S(Px_i) \, (i \geq 1, \, Px_0 = 0) \tag{8}$$

The amount of eggs laid by a female cohort on the $i$th day after adult emergence is the product of a fecundity of the cohort at temperature $T°C [F(T)]$, a proportion of eggs laid by the female at the $i$th day $[E(Px_i) - E(Px_{i-1})]$, and the survival proportion of the female at the $i$th day $[S(Px_i)]$. $Px_{i-1}$ and $Px_i$ are physiological ages obtained through the accumulation of the aging rates derived by the female aging model $[A(T)]$ to the $i$-$1$th and $i$th day, respectively.

**Two-phase OM.**  Egg production was simulated only in oviposition phase with the female that completed pre-oviposition phase (Fig 1B). The newly emerged females required a time (= pre-oviposition period) to develop reproductive organ and complete mating. A proportion of the female completing pre-oviposition at the $i$th day after adult emergence was predicted by subtracting the cumulative proportion at the $i$-$1$th day $[C_p(Px_{i-1})]$ from that at the $i$th day $[C_p(Px_i)]$, whose physiological ages, $Px_{i-1}$ and $Px_i$, were obtained by summating the developmental rates from PDRM $[D_p(T)]$. A proportion of the females entered oviposition phase the following day ($i+1$th) and simulated the daily proportion of eggs laid at the $j$th day after the onset of oviposition phase was calculated by subtracting a proportion of cumulative oviposition at the $j$-$1$th day $[E_o(Px_{j-1})]$ from that at the $j$th day $[E_o(Px_j)]$, whose physiological ages, $Px_{j-1}$ and $Px_j$, were calculated by ODRM $[D_o(T)]$. Therefore, two-phase OM is expressed as:

$$F(T) \cdot [E_o(Px_j) - E_o(Px_{j-1})] \cdot S(Px_i) \, (j \geq 1, \, i \geq 1, \, Px_0 = 0) \tag{9}$$

**Model simulation and validation.**  Both OMs, current OM and two-phase OM, were simulated to predict the daily egg production sequence laid by one female at a constant temperature ranging from 10°C to 40°C by 1°C accrual. Daily observed egg production at each temperature was examined in a chamber to validate models. The mean temperature of each observation was used for input. Simulation outputs of both models and observed eggs were compared by transforming a cumulative graph and Pearson correlation test. The PopModel 1.5 program [56] was used to simulate models as described above (Fig 6).

## Results

### Temperature effect on adult longevity and fecundity

Temperature affects the longevity and fecundity of the *B. dorsalis* adult (Table 1). Longevity in the adult female decreased from 116.8 d at 18.8 °C to 22.4 d at 34.9°C as the temperature increased over 18.8 °C, while it was likely stationary below 18.8°C except for a temporal decrease at 16.7 °C ($F_{6,118} = 27.69$, $P < 0.0001$). Male showed a similar response against temperature but decreased longevity except at 13.5°C ($F_{6,200} = 68.96$, P<0.0001). Eggs were laid

**Table 1. Longevity (days) and fecundity of *Bactrocera dorsalis* adults at various constant temperatures.**

| Temperature (°C) | Female | | | Male | |
|---|---|---|---|---|---|
| | n | Longevity (Mean ± SE[1]) | Fecundity (Mean ± SE) | n | Longevity (Mean ± SE) |
| 13.5 | 17 | 94.1 ± 10.72 (87.0) abc[2] | _[3] | 24 | 113.0 ± 5.68 (122.3) a |
| 16.7 | 20 | 53.5 ± 4.46 (54.5) cde | 63 ± 22.5 c | 21 | 51.4 ± 3.25 (53.8) cde |
| 18.8 | 18 | 116.8 ± 7.45 (107.0) ab | 922 ± 86.7 b | 29 | 79.6 ± 6.35 (76.5) bc |
| 23.5 | 20 | 78.5 ± 6.40 (71.0) bcd | 1,545 ± 160.6 a | 36 | 68.1 ± 4.03 (62.0) bcd |
| 28.1 | 16 | 43.3 ± 4.33 (39.0) def | 1,684 ± 131.9 a | 30 | 43.3 ± 2.08 (44.0) def |
| 32.0 | 17 | 42.1 ± 4.63 (37.8) def | 958 ± 86.9 b | 33 | 30.9 ± 1.79 (28.8) efg |
| 34.9 | 17 | 22.4 ± 1.81 (21.5) ef | 138 ± 34.2 c | 34 | 19.0 ± 0.59 (19.0) fg |

[1]Standard error.

[2]Means followed by the same letters in a column are not significantly different by HSD test at P = 0.05 (female longevity: $F_{6,118}$ = 27.69, P < 0.0001; Fecundity: $F_{6,118}$ = 55.57, P < 0.0001; male longevity: $F_{6,200}$ = 68.96, P < 0.0001).

[3]No eggs laid by the female. Numbers in the parentheses are median value of longevity at each temperature.

from 16.7°C to 34.9°C. No eggs were found nor a vigorous activity of adults observed at 13.5°C, inferring no mating behaviors. As temperature increased, fecundity increased sharply from 63 eggs at 16.7°C to 1,684 eggs at 28.1°C and then decreased rapidly ($F_{6,118}$ = 55.57, P < 0.0001). Both the pre-oviposition and oviposition period of the females varied across temperatures (Table 2). APOP became shorter from 38.1 d at 16.7°C to 4.6 d at 32.0°C, and then increased at 6.2 d at 34.9°C ($F_{5,86}$ = 130.25, P < 0.0001) and TPOP showed a similar tendency to APOP ranging from 91.2 d at 16.7°C to 21.5 d at 32.0°C. However, AOP increased rapidly from 103.1 d at 18.8°C to 23.6 d at 16.7°C and then decreased to 17.4 d at 34.9°C ($F_{5,86}$ = 32.09, P < 0.0001).

## Low development threshold (LDT) and thermal constant (TC)

A linear regression test on increasing aging rates over 18.8°C (Fig 2A and 2B) showed that both adult males and females had similar LDT near 16.0°C, but the TC of females was slightly higher than that of the male's (Table 3; Male: $F_{1,3}$ = 15.89, P < 0.0283, Female: $F_{1,3}$ = 15.02, P < 0.0304). LDTs and TCs of APOP, AOP, and TPOP were estimated onto the developmental rates and showed a linear relationship against temperature (Fig 4A and 4B, Table 3). TC of

**Table 2. Adult pre-oviposition period (APOP) (days), adult oviposition period (AOP), and total pre-oviposition period (TPOP) of *Bactrocera dorsalis*.**

| Temperature (°C) | No. female oviposited[1] | APOP (Mean ± SE[2]) | AOP (Mean ± SE) | TPOP[3] |
|---|---|---|---|---|
| 16.7 | 11 | 38.1 ± 3.06 a[4] | 23.6 ± 4.27 c | 91.2 |
| 18.8 | 17 | 16.2 ± 0.75 b | 103.1 ± 7.20 a | 49.1 |
| 23.5 | 20 | 9.0 ± 0.21 c | 69.5 ± 6.46 b | 31.1 |
| 28.1 | 16 | 5.4 ± 0.58 cd | 37.8 ± 4.24 c | 24.1 |
| 32.0 | 17 | 4.6 ± 0.15 d | 37.5 ± 4.66 c | 21.5 |
| 34.9 | 11 | 6.2 ± 0.5 cd | 17.4 ± 1.55 c | 29.9 |

[1]Female first laid a significant number (>5) of eggs consistently.

[2]Standard error.

[3]TPOP is the sum of APOP and the immature development period in a previous study [26].

[4]Means followed by the same letters in a column are not significantly different by HSD test at P = 0.05 (APOP: $F_{5,86}$ = 130.25, P < 0.0001; AOP: $F_{5,86}$ = 32.09, P < 0.0001).

**Table 3. Low developmental threshold (LDT) and thermal constant (TC) of adult longevity, adult pre-oviposition period (APOP), adult oviposition period (AOP), and total pre-oviposition period (TPOP) of *Bactrocera dorsalis*.**

| Stage | Equation[1] | $r^2$ | LDT (°C) | TC (Degree-day, DD) |
|---|---|---|---|---|
| Male adult longevity | 0.0023 X – 0.0363 | 0.84 | 15.7 | 433.4 |
| Female adult longevity | 0.0020 X – 0.0316 | 0.83 | 16.0 | 507.1 |
| APOP | 0.0126 X – 0.1806 | 0.99 | 14.3 | 79.2 |
| AOP | 0.0014 X – 0.0175 | 0.89 | 12.2 | 696.6 |
| TPOP | 0.0023 X – 0.0241 | 0.97 | 10.6 | 437.4 |

[1]Linear regression test (Male adult: $F_{1,3}$ = 15.89, P = 0.0283, Female adult: $F_{1,3}$ = 15.02, P = 0.0304, APOP: $F_{1,3}$ = 417.307, P = 0.00026, AOP: $F_{1,2}$ = 20.5065, P = 0.04547, TPOP: $F_{1,3}$ = 96.5145, P = 0.00224)

[2]Sum of both periods of the pre-oviposition and immature stage [26] according to the corresponding temperature.

AOP was nine times higher than that of APOP, although the LDT of AOP was approximately 2°C lower (APOP: $F_{1,3}$ = 417.30, P = 0.0002, AOP: $F_{1,2}$ = 20.50, P = 0.0454). The LDT of TPOP decreased to 10.6°C, close to LDT of the immature stage (Table 3; $F_{1,3}$ = 96.51, P = 0.0022).

### Non-linear model development

**Adult aging and survival model.** The exponential type Eq (2) used for an aging rate model in a moth [51] drew curves of best fit on the aging rates of both adult males and females, respectively (Fig 2A). Parameters of both aging models were estimated to be very close to each other (Male: $F_{2,3}$ = 205.76; P < 0.0001; $r^2$ = 0.993, Female: $F_{2,3}$ = 23.57; P = 0.0146; $r^2$ = 0.940) (Table 4). The minimum aging rate ($r_0$) was ≈ 0.01 and maximum temperature ($T_H$) was 52.0°C when the aging rate became ≈ 1.0. Survival at different temperatures was normalized in a probability distribution based on physiological age (Fig 2C and 2D). A modified two-parameter Weibull equation was well-fitted to adult survival (Male: $F_{1,147}$ = 3585.8; P < 0.0001; $r^2$ = 0.961, Female: $F_{1,109}$ = 3624.5; P < 0.0001; $r^2$ = 0.971) (Table 4). Parameter $\alpha$ was estimated as 1.0665 in males and 1.0921 in females, implying a physiological age when 50% of individuals survive, respectively.

**Fecundity model and ORM.** A quadratic equation was matched to the campaniform of the fecundity between 16.7°C and 34.9 °C (Fig 3A). Parameters, $T_L$ and $T_H$, were estimated as 16.3°C and 35.2°C, respectively ($F_{2,3}$ = 99.57; P = 0.0018; $r^2$ = 0.985) (Table 5). Daily egg

**Table 4. Estimated parameter values of the aging and survival model of the *Bactrocera dorsalis* adult.**

| Model[1] | Adult | Parameter | Value | P | $r^2$ |
|---|---|---|---|---|---|
| Aging model | Male | $r_0$ | 0.0093 | 0.01198 | 0.99 |
| | | $T_H$ | 52.1197 | 0.0002 | |
| | | $c$ | 5.4531 | 0.00567 | |
| | Female | $r_0$ | 0.0096 | 0.07643 | 0.94 |
| | | $T_H$ | 51.4995 | 0.00427 | |
| | | $c$ | 4.8952 | 0.08800 | |
| Age-specific survival rate model | Male | $\alpha$ | 1.0665 | <0.0001 | 0.96 |
| | | $\beta$ | 3.7744 | <0.0001 | |
| | Female | $\alpha$ | 1.0921 | <0.0001 | 0.97 |
| | | $\beta$ | 2.9417 | <0.0001 | |

[1]A modified function [51] was used for the adult aging model (Male: $F_{2,3}$ = 205.76; P < 0.0001, Female: $F_{2,3}$ = 23.57; P = 0.0146) and a modified model of the two-parameter Weibull equation [52] was applied to the survival rate model (Male: $F_{1,147}$ = 3585.8; P < 0.0001, Female: $F_{1,109}$ = 3624.5; P < 0.0001)

**Table 5. Estimated parameter values of the component model of the current oviposition model and two-phase oviposition model.**

| Model[1] | Parameter | Value | P | $r^2$ |
|---|---|---|---|---|
| Total fecundity model | $\alpha$ | 19.3991 | 0.0008 | 0.985 |
| | $T_L$ | 16.3221 | 0.0000 | |
| | $T_H$ | 35.2481 | <0.0001 | |
| Age-specific cumulative oviposition rate model (ORM) | $\gamma$ | 0.1020 | <0.0001 | 0.968 |
| | $\eta$ | 0.2495 | <0.0001 | |
| | $\beta$ | 1.2024 | <0.0001 | |
| Pre-oviposition development rate model | $\alpha$ | $2.8177 \times 10^{-4}$ | 0.01115 | 0.996 |
| (PDRM) | $T_L$ | 13.3204 | 0.00655 | |
| | $T_H$ | 35.1973 | 0.00008 | |
| | $m$ | 4.4668 | 0.10455 | |
| Pre-oviposition complete distribution model | $\gamma$ | 0.4852 | 0.00559 | 0.892 |
| (PCDM) | $\eta$ | 0.4530 | 0.01285 | |
| | $\beta$ | 2.3140 | 0.3536 | |
| Oviposition development rate model | $a$ | 0.0393 | 0.01211 | 0.989 |
| (ODRM) | $b$ | $1.99645 \times 10^{-17}$ | 0.01895 | |
| | $c$ | -12.7215 | 0.03574 | |
| | $d$ | 871175 | 0.01537 | |
| Age-specific cumulative oviposition rate model of oviposition period (OORM) | $\alpha$ | 0.2401 | <0.0001 | 0.981 |
| | $\beta$ | 1.0459 | <0.0001 | |

[1]The quadratic equation was applied to the total fecundity model ($F_{2,3}$ = 99.57; P = 0.0018); the three-parameter Weibull function [54] was applied to ORM ($F_{2,367}$ = 5486.6; P < 0.0001) and PCDM ($F_{2,40}$ = 165.04; P < 0.0001); the Briere 2 model [55] was applied to PDRM ($F_{3,2}$ = 168.02; P = 0.00592); a best-fit equation was selected for ODRM ($F_{3,2}$ = 59.64; P = 0.0165) using the Table Curve 2D program [49] and a modified model of the two-parameter Weibull equation [52] was applied to OORM ($F_{1,389}$ = 20583.1; P < 0.0001).

production laid by the female at different temperature was translated in an age-specific cumulative probability distribution (Fig 3B). The curve of the three-parameter Weibull function described the distribution well ($F_{2,367}$ = 5486.6; $P < 0.0001$; $r^2$ = 0.968) (Table 5). Parameter $\gamma$ was 0.102 implying the female physiological age where they can start laying eggs and 50% of eggs were laid when the female was aged 0.3515 ($\gamma+\eta$), physiologically.

**Component models of two-phase OM: PDRM and PCDM in pre-oviposition and ODRM and OORM in oviposition.** Pre-oviposition development of the female was well described with PDRM ($F_{3,2}$ = 168.027; P = 0.00592; $r^2$ = 0.996) (Fig 4A, Table 5). Parameters, $T_L$ and $T_H$, were estimated as 13.3°C and 35.2°C, respectively. The cumulative probability distribution of the females that completed pre-oviposition was slightly scattered, but the estimated curve of PCDM was well-fitted to the distribution ($F_{1,389}$ = 20583.1; $P < 0.0001$; $r^2$ = 0.892) (Fig 4C, Table 5). Parameter $\gamma$ was 0.4852 for the first occurrence of the females who completed pre-oviposition phase, and 50% of the females completed pre-oviposition phase when aged 0.9382 ($\gamma+\eta$). In oviposition phase, the 'U' shape of the female development rate was well-fitted to the curve of ODRM ($F_{3,2}$ = 59.6469; P = 0.01653; $r^2$ = 0.989) (Fig 4B, Table 5). Cumulative oviposition probability distribution was well-described with OORM ($F_{1,389}$ = 20583.1; $P < 0.0001$; $r^2$ = 0.981) (Fig 4D, Table 5). In total, 50% fecundity of eggs is laid at a physiological age of 0.2401 (parameter $\alpha$).

## Model simulation and comparison with observation

Both OMs, current OM and two-phase OM, predicted a daily egg production sequence laid by one female in a temperature range from 10°C to 40°C (Fig 5). Eggs were laid only between

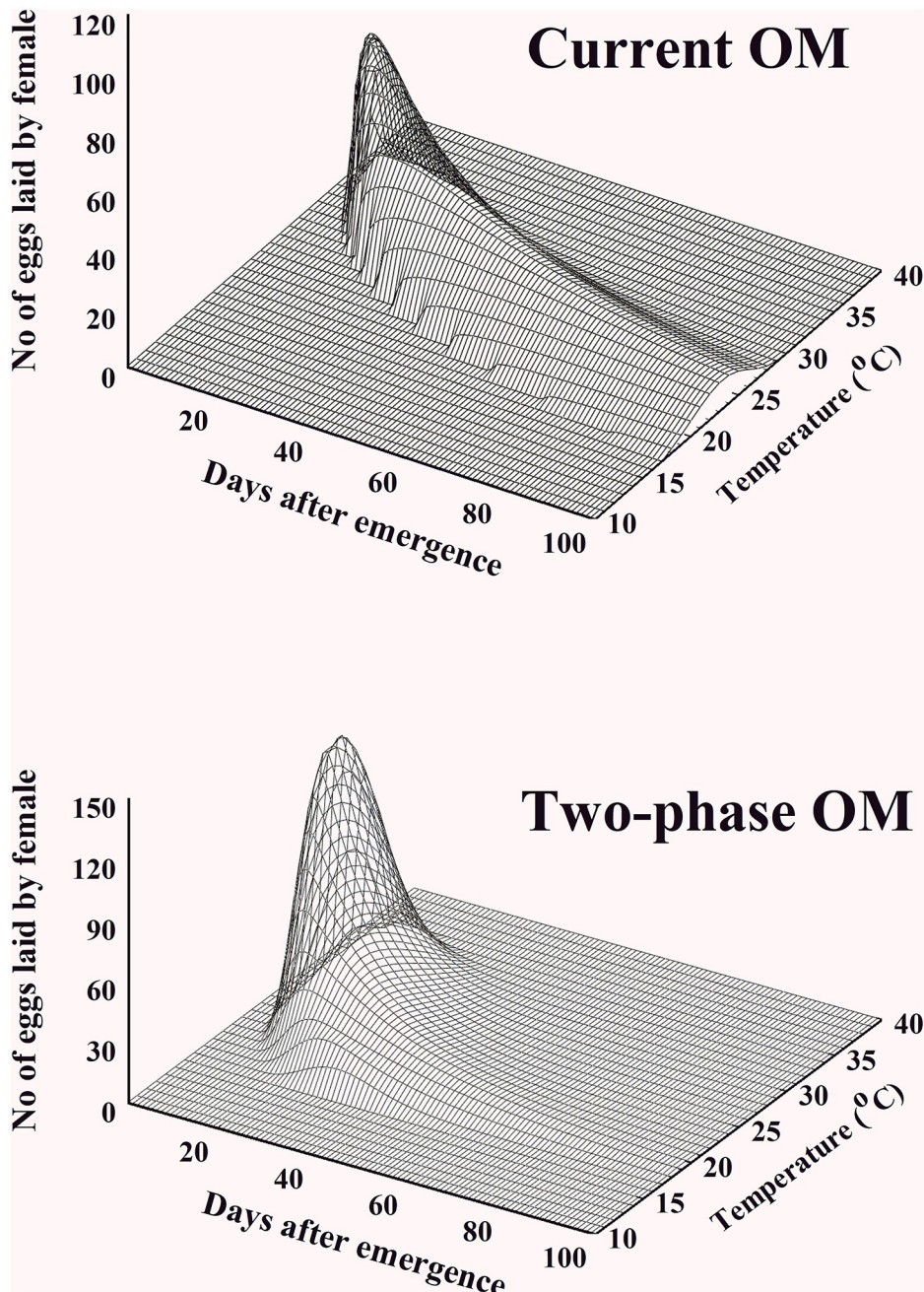

**Fig 5. Daily egg production sequences of the *Bactrocera dorsalis* female predicted by the oviposition model (A) and two-phase oviposition model (B) at a certain temperature from 10˚C to 40˚C after the female adult emerged.**

17˚C and 35˚C in both models. However, the egg production of current OM lasted longer than that of two-phase OM. For example, in the case of two-phase OM, over 10 eggs were predicted at the 4th to 50th days after adult emergence, but current OM predicted this up to the 102nd day. Additionally, the output of two-phase OM better matched the observed egg production in all observations than the output of current OM as well as the pre-oviposition period in 16.7˚C, 18.8˚C, and 34.9˚C (Fig 6, Table 6).

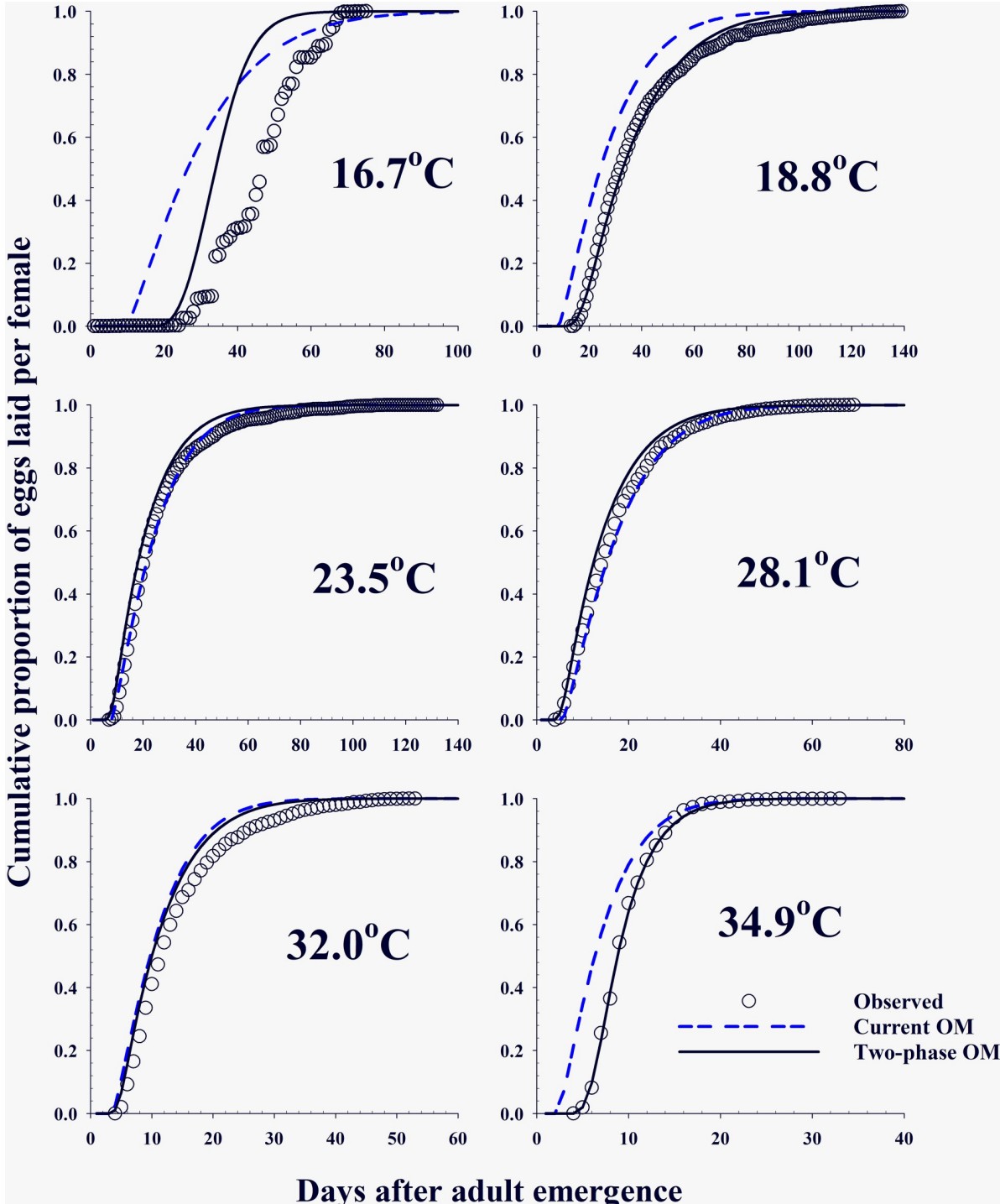

**Fig 6. Comparison of simulation outputs of two models: Current oviposition model and two-phase oviposition model with the observed egg rate (observed) at each different temperature.**

**Table 6. Comparison of outputs from both Oviposition Models (OMs), current OM and two-phase OM with observed eggs laid by *Bactrocera dorsalis* females at different constant temperatures.**

| Temperature (˚C) | Pearson correlation coefficient ($r$) | |
|:---:|:---:|:---:|
| | **Current OM** | **Two-phase OM** |
| 16.7 | 0.863***[1] | 0.919*** |
| 18.8 | 0.993*** | 0.998*** |
| 23.5 | 0.997*** | 0.997*** |
| 28.1 | 0.999*** | 0.996*** |
| 32.0 | 0.993*** | 0.995*** |
| 34.9 | 0.988*** | 0.999*** |

[1]Highly highly significant (P < 0.0001).

## Discussion

Temperature influenced the longevity of adults as it affected the development periods during immature stage [38, 57]. The aging process increases in terms of speed as temperature increases for adults. The longevity of the *B. dorsalis* adults becomes shorter (= ages faster) in increasing temperature conditions, and adults cannot live even a day over 52˚C, which is the estimated maximum temperature ($T_H$) of the aging model for both sexes (Fig 2A and 2B, Tables 1 and 4). On the other hand, the aging rate of both sexes slows as temperature decreases, and is stationary below 18.8˚C, implying a minimum aging rate ($\approx 0.01$) per day for the adult, regardless of temperature. Given the estimated minimum aging rate ($r_0$) of the aging model, we expect that the average of the innate longevity ($1/r_0$) is 103 days for females and 107 days for males of *B. dorsalis*. The longevity of *B. dorsalis* observed by Yang et al [58] was longer than those observed in the present study. It is inferred that the difference between the two populations may be influenced by the food resources of the larvae and adults.

Reproduction is an important biological process for maintaining the population size and to transmit the genetic information of a species. Mating is a prerequisite for reproduction, except for parthenogenesis, but harsh environments like cold or hot temperatures hinder both sexes from mating with each other. Although we did not investigate their mating behavior and egg hatchability, we deduced a possible optimal temperature range for reproduction as no eggs were found at 13.5˚C, where both *B. dorsalis* adults had survived for about 100 days, while eggs were observed in the other temperature treatments in this study (Table 1). Considering some of the eggs might be unfertilized, this observation infers that a suitable temperature for successful mating must be within a temperature range between 16.7˚C and 34.9˚C. Adults can emerge when the ground temperature is above 16˚C and the optimum temperature is 22˚C [15]. According to field occurrences, the optimal temperature for adult activity may range between 18~32˚C [24, 59]. Additionally, there is a pre-copulation period and a specific period of mating time. For example, mating behavior starts 10 days after emergence and the mating rate reaches 50% at the 29th day and 80% at the 51st day, which is related to ovary development [60].

*Bactrocera dorsalis* females show a temperature-dependent development relationship for adult pre-oviposition period (APOP) and adult oviposition period (AOP) (Fig 4A and 4B, Table 2). The optimal oviposition temperature range is likely between 18.8˚C and 32.0˚C because aging rates, reciprocals of the oviposition period, showed a proportional relationship to temperature only while it increased rapidly below or above the range. Fecundity did not show a linear relationship but is close to a symmetric campanula shape with a higher amount of eggs at 23.5˚C and 28.8˚C and the lowest at 16.7˚C and 34.9˚C at each side (Fig 3A, Table 5). The highest fecundity for different species occurred from 20˚C to 25.2˚C: 20˚C for

*C. sasakii* [33], 20.3˚C for *R. padi* [45], 21˚C for *O. sulcatus* [35], 24˚C for *N. californicus* [40], 25˚C for *R pedestris* [37], *T. basalis* [42] and *E. kuehniella* [44], and 25.2˚C for *C. medinalis* [41]. Parameters, $T_L$ and $T_H$, of the fecundity model were estimated as 16.3˚C and 35.2˚C, implying low and high temperature limitation for reproduction, respectively (Table 5). Daily egg production was dependent on temperature and most eggs were laid before the female aged to her half-life in that temperature (Fig 6). Cumulative egg production curves (Figs 3B, 4C and 4B) are a type of temperature-effect-removed distribution that depicts innate egg production based on the female's physiological age.

Many studies have modeled a linear relationship between insect development and temperatures [61,62]. The low development threshold (LDT) and thermal constant (TC) of adults were estimated as 15.7˚C and 433.4 DD for males and 16.0˚C and 507.1 DD for females. TPOP is the generation time from the egg to when the fully grown adult lays its egg and whose LDT and TC were estimated as 10.6˚C and 437.4 DD, which is similar to the LDT and TC of the Chinese population: 10.7˚C and 501.7 DD [14]; and 12.2˚C and 334.4 DD [21]. Although the degree-day model is simple for calculating their occurrence, it could not predict daily reproduction or even adult survival.

The logical structure of the current model has no room to adopt a concept for the pre-oviposition period. For instance, *B. dorsalis* adult can live several months and have a considerable APOP before laying eggs. Therefore, APOP is long enough to affect female mortality and mating. Nonetheless, there is individual variance in the period. Furthermore, the current OM forced females complete APOP when aged 0.1020, as indicated by parameter $\gamma$ of ORM. However, two-phase OM developed for *B. dorsalis* bypassed this problem and showed improved and more realistic simulation outputs according to the observations (Figs 5 and 6). The detailed physiological mechanisms underlying the aging of insect merit further research.

We showed a temperature effect on the longevity and fecundity of *B. dorsalis* adults reared with an artificial diet, although *B. dorsalis* females had a host preference for oviposition and their reproduction was affected by the host plants they had grown [30]. We improved the current OM using a two-phase OM and precisely estimated egg production under various temperatures, especially lower temperature conditions. Furthermore, two-phase OM will be useful to predict egg production of insects which have long APOP period. The actual egg populations from the field were not validated in this study but we can estimate the egg phenology in *B. dorsalis* and predict the lower and higher temperature thresholds of fecundity. In a previous study, we developed a temperature dependent development model for *B. dorsalis* [26]. A population model structured by temperature dependent development and the new OM structure is helpful for predicting the egg production of *B. dorsalis* under field conditions and may be useful for understanding the population dynamics of this species. Additional studies on the stable and fluctuating temperature effects of the biological traits of *B. dorsalis*. are still needed.

## Supporting information

**S1 Table. The estimated development rate of *Bactrocera dorsalis* female at various constant temperatures.**
(DOCX)

**S2 Table. The estimated development rate of *Bactrocera dorsalis* male at various constant temperatures.**
(DOCX)

**S3 Table. The physiological age and survival probability of *Bactrocera dorsalis* female at various constant temperatures.**
(DOCX)

**S4 Table. The estimated survival probability of *Bactrocera dorsalis* female.**
(DOCX)

**S5 Table. The physiological age and survival probability of *Bactrocera dorsalis* male at various constant temperatures.**
(DOCX)

**S6 Table. The estimated survival probability of *Bactrocera dorsalis* male.**
(DOCX)

**S7 Table. The estimated total fecundity of *Bactrocera dorsalis* at various constant temperatures.**
(DOCX)

**S8 Table. The physiological age and cumulative proportion of egg production of *Bactrocera dorsalis* at various constant temperatures.**
(DOCX)

**S9 Table. The estimated cumulative proportion of egg production of *Bactrocera dorsalis*.**
(DOCX)

**S10 Table. The estimated development rate of *Bactrocera dorsalis* female in the pre-oviposition at various constant temperatures.**
(DOCX)

**S11 Table. The cumulative proportion for completing pre-oviposition of *Bactrocera dorsalis* at various constant temperatures.**
(DOCX)

**S12 Table. The estimated cumulative proportion of completing pre-oviposition period of *Bactrocera dorsalis* female.**
(DOCX)

**S13 Table. The estimated development rate of *Bactrocera dorsalis* female in the oviposition at various constant temperatures.**
(DOCX)

**S14 Table. The cumulative proportion of oviposition of *Bactrocera dorsalis* female at various constant temperatures.**
(DOCX)

**S15 Table. The estimated cumulative proportion of oviposition of *Bactrocera dorsalis* female.**
(DOCX)

## Acknowledgments

Dr. Kyung San Choi and Shaw-Yhi Hwang were received funds from Rural Development Administration and the study was carried out with the cooperative research project (Project number: PJ01207501) between National Chung Hsing University (NCHU), Taiwan and

National Institute of Horticultural and Herbal Science (NIHHS), Rural Development Administration (RDA), Republic of Korea.

## Author Contributions

**Conceptualization:** Kyung San Choi, Shaw-Yhi Hwang, Yu-Bing Huang, Jeong Joon Ahn.

**Data curation:** Kyung San Choi, Ana Clariza Samayoa, Jeong Joon Ahn.

**Formal analysis:** Kyung San Choi, Jeong Joon Ahn.

**Funding acquisition:** Kyung San Choi, Shaw-Yhi Hwang.

**Investigation:** Kyung San Choi, Ana Clariza Samayoa, Jeong Joon Ahn.

**Methodology:** Kyung San Choi, Ana Clariza Samayoa, Yu-Bing Huang, Jeong Joon Ahn.

**Project administration:** Kyung San Choi, Shaw-Yhi Hwang, Jeong Joon Ahn.

**Resources:** Kyung San Choi, Shaw-Yhi Hwang, Yu-Bing Huang, Jeong Joon Ahn.

**Software:** Kyung San Choi, Ana Clariza Samayoa, Jeong Joon Ahn.

**Supervision:** Kyung San Choi, Shaw-Yhi Hwang, Jeong Joon Ahn.

**Validation:** Kyung San Choi, Shaw-Yhi Hwang, Jeong Joon Ahn.

**Visualization:** Kyung San Choi, Jeong Joon Ahn.

**Writing – original draft:** Kyung San Choi, Jeong Joon Ahn.

**Writing – review & editing:** Kyung San Choi, Ana Clariza Samayoa, Shaw-Yhi Hwang, Yu-Bing Huang, Jeong Joon Ahn.

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
