## [Decision Letter · Decision Letter 0]

3 Apr 2020

PONE-D-20-03016

Thermal effect on the fecundity and longevity of Bactrocera dorsalis adults and their improved oviposition model

PLOS ONE

Dear Dr. Ahn,

Thank you for submitting your manuscript to PLOS ONE. After careful consideration, we feel that it has merit but does not fully meet PLOS ONE’s publication criteria as it currently stands. Therefore, we invite you to submit a revised version of the manuscript that addresses the points raised during the review process, these include:

1) reduce/simplify the abbreviations and technical jargon

2) provide greater clarification on the approach (and the need for the approach) as well as the potential underlying phyiology

3) answer specific questions from Reviewer 2 re the experimental conditions used (see attached pdf)

4) address how variable temperature conditions have affected other species

We would appreciate receiving your revised manuscript by May 17 2020 11:59PM. To enhance the reproducibility of your results, we recommend that if applicable you deposit your laboratory protocols in protocols.io, where a protocol can be assigned its own identifier (DOI) such that it can be cited independently in the future. For instructions see: http://journals.plos.org/plosone/s/submission-guidelines#loc-laboratory-protocols

We look forward to receiving your revised manuscript.

Kind regards,

J Joe Hull, Ph.D.

Academic Editor

PLOS ONE

Journal Requirements:

2. Please include your tables as part of your main manuscript and remove the individual files. Please note that supplementary tables (should remain/ be uploaded) as separate "supporting information" files

Reviewers' comments:

Reviewer's Responses to Questions

**Comments to the Author**

1. Is the manuscript technically sound, and do the data support the conclusions?

Reviewer #1: Yes

Reviewer #2: Partly

2. Has the statistical analysis been performed appropriately and rigorously? 

Reviewer #1: Yes

Reviewer #2: Yes

3. Have the authors made all data underlying the findings in their manuscript fully available?

Reviewer #1: Yes

Reviewer #2: Yes

4. Is the manuscript presented in an intelligible fashion and written in standard English?

Reviewer #1: Yes

Reviewer #2: Yes

5. Review Comments to the Author

Reviewer #1: In this manuscript, the authors set out to empirically measure the effects of 7 constant temperatures on the development time, longevity, and reproductive output of Bactrocra dorsalis. Further, it seeks to develop a two-step model for population growth that includes a pre-oviposition adult period (which seems to be lacking in the literature). Overall, I think the science is sound and the model seems fine. I do have some comments that I hope may improve the manuscript.

Comments:

There is an abrupt transition between line 65 and 66. There seems to be a bit of a leap in logic. There should be more of a justification of how your data and approach are different than slightly a modified Kim and Lee model.

Aging as rate, when measured only as a function of 1/longevity, seems a bit strange. For the model, this rate seems ok but for the description of the physiology of the organisms at different temperatures, I do not think it is correct. That is to say, you think a fly on day 22 at a high temperature has experienced the same amount of “aging” (teleomere shortening, accumulation of physiological damage, etc) as a fly at 166 days at a cold temperature? Without data to support that, I think it might not be the correct term for describing what is going on.

Generally, there is a mix of abbreviations and spelled out terms which becomes confusing. Especially in the model description.

The discussion cold benefit from some discussion of fluctuating temperatures versus constant temperatures.

Reviewer #2: This paper measures the longevity and fecundity of Bactrocerca dorsalis at various constant temperatures to predict oviposition. The new model takes into account the long pre-ovipositional period of the species.

While the Introduction is succinct, the need for an improved oviposition model can be emphasized. Otherwise, the reader does not know what has been done, and why this new model is an improvement. Many of the points in paragraph 363-374 would be relevant in the Introduction.

Many acronyms are used, and the reader has to keep track of these. Perhaps a more limited set of acronyms could be used. POP and OP are the main acronyms. These terms can be preceded by the more descriptive term such as Adult POP, Adult OP, Total POP.

Methods

It is not clear whether flies were set up all at once, or was a 1 female: 2 male set up for each treatment over several days.

Table 2 shows the sample size to be 11-20 females per temperature. Eleven replicates seems on the low side.

Line 106 – Total pre-ovipositional period include data from another paper. What were the immature develop periods, this might be shown in the Table 2 as another column. Also, if this other data is added, were the conditions of measuring development similar?

Fixed temperatures are used in models, while this is often done, what are the typical temperature fluctuations in a day where this fly is found. It should be mentioned that development could occur slower or faster with variable temperatures. Other species have been monitored with variable temperatures, what did they find with variable temperatures with the same average as fixed temperatures?

6. PLOS authors have the option to publish the peer review history of their article (what does this mean?). If published, this will include your full peer review and any attached files.

Reviewer #1: No

Reviewer #2: No

---

## [Author Response · Author response to Decision Letter 0]

1 May 2020

Response to Reviewers’ Comments

All authors really appreciate two anonymous reviewers for valuable comments on our manuscript.

Response to Editor’s Comments

Thank you for ….these include:

[RE] The authors appreciate your positive review. We revised some parts of manuscript and tables.

1) reduce/simplify the abbreviations and technical jargon

[RE] We revised it as suggested. Abbreviation “POP” and “OP” stand for pre-oviposition phase and oviposition phase. Those are not general terms used in this kind of study but a term used to indicate separate phase in two-phase oviposition model of this study. We removed those two abbreviations for reducing any confusion in the manuscript. Other similar abbreviations, i.e. APOP, AOP, and TPOP, used in this manuscript. We introduce these abbreviations for compatibility with other studies because these abbreviations are often found other articles so that we leave them.

Examples of recent article using APOP, AOP and TPOP

Shi M-Z, Li J-Y, Ding B, Fu J-W, Zheng L-Z, Chi H. Indirect effect of elevated CO2 on population parameters and growth of Agasicles hygrophila (Coleoptera: Chrysomelidae), a biological agent of alligatorweed (Amaranthaceae). J Econ Entomol. 2019;112: 1120-1129.

Chen G-M, Chi H, Wang R-C, Wang Y-P, Xu Y-Y, Li X-D, Yin P, Zheng F-Q. 2018. Demography and uncertainty of population growth of Conogethes punctiferalis (Lepidoptera: Crambidae) reared on five host plants with discussion on some life history statistics. J Econ Entomol. 2018; 111: 2143-2152.

Liu J, Huang W, Chi H, Wang C, Hua H, Wu G. 2017. Effects of elevated CO2 on the fitness and potential population damage of Helicoverpa armigera based on two-sex life table. Scientific Reports. 2017;7: 1119.

Chen Q, Li N, Wang X, Ma L, Huang J-B, Haung G-U. Age-stage, two-sex life table of Parapoynx crisonalis (Lepidoptera: Pyralidae) at different temperatures. PLoS ONE. 2017:12; e0173380.

Oviposition models suggested by this study are composed of several unit models and have complicated structure. If reader has no background about this kind of works they may be confused the long and similar name. In order to minimize such confusion, we introduce simple abbreviations, i.e. ORM, PDRM, PCDM, ODRM, OORM, which would be better than their own name like Age-specific cumulative oviposition rate model. For readers, it is more convenient to use abbreviation in the manuscript. 

2) provide greater clarification on the approach (and the need for the approach) as well as the potential underlying physiology

[RE] We thank for pointing it out. Please refer to line 393-394. This study is a part of series to build the phenology model of Bactrocera dorsalis to understand their occurrence in field and help protect their damage by predicting population dynamics. Although we studied the effects of temperature on the development of their immature stage, they spend their immature life inside fruit and soil and also immature period is relatively short (a couple of weeks). However, adult can survive for several months depending on temperature conditions. Therefore, atmospheric temperature can affect their life history like mating, oviposition, damage to fruit, spread and distribution. In this study, we examined longevity of male and female, and oviposition of female under different temperature conditions. Furthermore, we developed an improved oviposition model, two-phase oviposition model for the first time. Maintaining genome integrity is one of important factors in longevity and cell viability. Telomere length regulation is an important aspect of cell maintenance in eukaryotes (Hasty et al. 2003; Rodier e al. 2005). Various molecular mechanisms are likely to cause the specific telomere dynamics, including cell division, oxidative stress and telomerase activity (Jemiellty et al. 2007). Additional studies on the relationship between demographic and physiological approach for understanding and predicting the aging of insect are still needed. 

Hasty P, Campisi J, Hoeijmakers J, van Steeg H, Vijg J. Aging and genome maintenance: lessons from the mouse? Science. 2003;299: 1355-1359.

Jemiellty S, Kimura M, Parker KM, Parker JD, Cao X, Aviv A, Keller L. Short telomeres in short-lived males: what are the molecular and evolutionary causes? Ageing Cell. 2007;6: 225-233. 

Rodier F, Kim S-H, Niijar T, Yaswen P, Campisi J. Cancer and aging: the importance of telomeres in genome maintenance. The international Journal of Biochemistry & Cell Biology. 2005;37: 977-990.

3) answer specific questions from Reviewer 2 re the experimental conditions used (see attached pdf)

[RE] We respect the comments. We give the answers from Reviewer 2’s questions.

Experimental Condition

Question 1: It is not clear whether flies were set up all at once, or was a 1 female: 2 males set up for each treatment over several days.

[RE] We have three growth chambers for experiment. When we start the experiment at each temperature treatment we set up all replication within a day with one female and two males. 

Question 2: Table 2 shows the sample size to be 11-20 females per temperature. Eleven replicates seem on the low side

[RE] We thank for pointing it out. The experiment started with 18 or 20 cages including one female and two males per cage. When we check the number of eggs and survivorship of male and female in each cage, adult can escape from the cage. We excluded all examined data related with the escaped adults. Therefore, in table 1, “n” means the number of female that we examined successfully during the experiment without any incident that lost female in a cage. In table2, “no. female oviposited” means the number of female oviposited from female of table 1.

Question 3: Line 106 – Total pre-ovipositional period include data from another paper. What were the immature develop periods, this might be shown in the Table 2 as another column. Also, if this other data is added, were the conditions of measuring development similar?

[RE] We thank for pointing it out. Total pre-ovipositional period (TPOP) was not examined from this study. TPOP deduced by summing APOP and development period of immature stage. We provide TPOP values for a reader who will need from this and our previous studies. The recorded temperature difference between this and previous study are less than 0.5°C except 20°C (0.7°C).

4) address how variable temperature conditions have affected other species

[RE] We respect the comment. Please refer to line 405-406. Insects are ectotherm and the effect of temperature on the development, longevity, fecundity and distribution of ectotherms has been well studied. Effects of temperature on those performances are different from species and life stages even though it is a same temperature. Specially, the current oviposition model has been developed for various insects and mites such as Carposina sasakii Matsumura (Kim and Lee 2003), Tetranychus urticae Koch (Kim and Lee 2003), Otiorhynchus sulcatus (F) (Son and Lewis 2005), Scotinophara lurida (Burmeister) (Kim and Lee 2008), Riptortus pedestris (Thunberg) (Kim et al. 2009, Ahn et al. 2019), Neoseiulus californicus (McGregor) (Kim et al. 2013), Cnaphalocrocis medinalis Guenée (Park et al. 2014), Trissolcus basalis Wholaston (Forouzan et al., 2015), Ephestia kuehniella (Zeller) (Pakyari et al. 2016), and Rhopalosiphum padi (Linnaeus) (Park et al. 2017). The temperature range of the highest total fecundity for each species was from 20°C to 25.2°C: 20°C for C. sasakii, 20.3°C for R. padi, 21°C for O. sulcatus, 24°C for N. californicus, 25°C for R pedestris, T. basalis and E. kuehniella, and 25.2°C for C. medinalis. The oviposition model of C. sasakii was incorporated into the population model of an orchard system for establishing management strategies (Kim and Lee 2010). 

Kim D-S, Lee J-H. Oviposition model of Carposina sasakii (Lepidoptera: Carposiniae). Ecol Model. 2003;162: 145-153. https://doi.org/10.1016/s0304-3800(02)00402-7.

Kim D-S, Lee J-H. Oviposition model of overwintered adult Tetranychus urticae (Acari: Tetranychidae) and mite phenology on the ground cover in apple orchards. Experimental and Applied Acarology. 2003;31: 191-208. 

Son Y, Lewis EE. Effects of temperature on the reproductive life history of the black vine weevil, Otiorhynchus sulcatus. Entomol Exp Appl. 2005;114: 15-24.

Kim H, Lee J-H. Phenology simulation model of Scotinophara lurida (Hemiptera: Pentatomidae). Environ Entomol. 2008;37: 660-669.

Kim H, Baek S, Kim S, Lee S-Y, Lee J-H. Temperature-dependent development and oviposition model of Riptortus clavatus (Thunberg) (Hemiptera: Alydidae). Appl Entomol Zool. 2009;44: 515-523.

Ahn JJ, Choi KS, Koh S. Effects of temperature on the development, fecundity and life table parameters of Riptortus pedestris (Hemiptera: Alydidae). Appl Entomol Zool. 2019;54: 63-74. https://doi.org/10.1007/s13355-018-0593-5.

Kim T, Ahn JJ, Lee J-H. Age- and temperature-dependent oviposition model of Neoseiulus californicus (McGregor) (Acari: Phytoseiidae) with Tetranychus urticae as prey. J Appl Entomol. 2013;137: 282-288.

Park H-H, Park C-G, Ahn JJ. Oviposition model of Cnaphalocrocis medinalis Guenee (Lepidoptera: Pyralidae). J Asia-Pac Entomol. 2014;17: 781-786.

Forouzan M, Shirazi J, Safaralizadeh MH, Safavi SA, Rezaei M. Oviposition model of Trissolcus basalis Wholaston (Hym.: Scelionidae) on sunn pest eggs. J Agr Sci Tech. 2015;17: 551-560.

Pakyari H, Amir-Maafi M, Moghadamfar Z. Oviposition model of Ephestia kuehniella (Lepidoptera: Pyralidae). J Econo Entomol. 2016;109: 2069-2073.

Park C-G, Choi B-R, Cho JR, Kim J-H, Ahn JJ. Thermal effects on the development, fecundity and life table parameters of Rhopalosiphum padi (Linnaeus) (Hemiptera: Aphididae) on barley. J Asia-Pac Entomol. 2017;20: 767-775.

Kim D-S, Lee J-H. A population model for the peach fruit moth, Carposina sasakii Matsumura (Lepidoptera: Carposinidae), in a Korean orchard system. Ecol Model. 2010;221: 268-280.

Response to Reviewer 1’s Comments

In this manuscript …improve the manuscript.

[RE] The authors appreciate your positive review. We revised some parts of manuscript. 

There is an abrupt transition… a modified Kim and Lee model.

[RE] We thank for pointing it out. Please refer to line 66-80. We change the order of figures for explaining the modified model. 

Aging as rate …..what is going on. 

[RE] We respect the comment. Please refer to line 393-394. In 1980s, physiological age concept was introduced to solve the problem as you mentioned the flies lived the same days under different temperature conditions. Aging model produce a mathematical function how insect takes old under different temperature conditions. Physiological age, a sum of those temperature-dependent rate values that female has experienced after adult emergence, was applied to represent age of the female cohort according to mathematical logic developed by Curry and Feldman (1987). Aging model is developed by 1/ mean longevity and it produce one static value. Such distribution in survival and oviposition, even though females has same age, are obtained by transforming survival and oviposition according to the mean longevity. Therefore, different biological process caused by factors that are not related with temperature were incorporated in age-specific survival rate models and age-specific cumulative oviposition model. This kind of model process has been developed regarding such physiological variation that reviewer indicated. Exactly ageing model produce a mean age of the cohort at a day after adult emergence. The variation of age-specific survival and cumulative oviposition distribution describe the variation (Fig 2C, 2D, 3B, 4C and 4D). 

Maintaining genome integrity is one of important factors in longevity and cell viability. Telomere length regulation is an important aspect of cell maintenance in eukaryotes (Hasty et al. 2003; Rodier e al. 2005). Various molecular mechanisms are likely to cause the specific telomere dynamics, including cell division, oxidative stress and telomerase activity (Jemiellty et al. 2007). Additional studies on the relationship between demographic and physiological approach for understanding and predicting the aging of insect are still needed. 

Hasty P, Campisi J, Hoeijmakers J, van Steeg H, Vijg J. Aging and genome maintenance: lessons from the mouse? Science. 2003;299: 1355-1359.

Jemiellty S, Kimura M, Parker KM, Parker JD, Cao X, Aviv A, Keller L. Short telomeres in short-lived males: what are the molecular and evolutionary causes? Ageing Cell. 2007;6: 225-233. 

Rodier F, Kim S-H, Niijar T, Yaswen P, Campisi J. Cancer and aging: the importance of telomeres in genome maintenance. Int J Biochem Cell B. 2005;37: 977-990.

Generally, there is mix …..in the model description.

[RE] We thank for pointing it out. We revised it as suggested. Abbreviation “POP” and “OP” stand for pre-oviposition phase and oviposition phase. Those are not general terms used in this kind of study but a term used to indicate separate phase in two-phase oviposition model of this study. We removed those two abbreviations for reducing any confusion in the manuscript. Other similar abbreviations, i.e. APOP, AOP, and TPOP, used in this manuscript. We introduce these abbreviations for compatibility with other studies because these abbreviations are often found other articles so that we leave them.

Examples of recent article using APOP, AOP and TPOP

Shi M-Z, Li J-Y, Ding B, Fu J-W, Zheng L-Z, Chi H. Indirect effect of elevated CO2 on population parameters and growth of Agasicles hygrophila (Coleoptera: Chrysomelidae), a biological agent of alligatorweed (Amaranthaceae). J Econ Entomol. 2019;112: 1120-1129.

Chen G-M, Chi H, Wang R-C, Wang Y-P, Xu Y-Y, Li X-D, Yin P, Zheng F-Q. 2018. Demography and uncertainty of population growth of Conogethes punctiferalis (Lepidoptera: Crambidae) reared on five host plants with discussion on some life history statistics. J Econ Entomol. 2018; 111: 2143-2152.

Liu J, Huang W, Chi H, Wang C, Hua H, Wu G. 2017. Effects of elevated CO2 on the fitness and potential population damage of Helicoverpa armigera based on two-sex life table. Scientific Reports. 2017;7: 1119.

Chen Q, Li N, Wang X, Ma L, Huang J-B, Haung G-U. Age-stage, two-sex life table of Parapoynx crisonalis (Lepidoptera: Pyralidae) at different temperatures. PLoS ONE. 2017:12; e0173380.

Oviposition models suggested by this study are composed of several unit models and have complicated structure. If reader has no background about this kind of works they may be confused the long and similar name. In order to minimize such confusion, we introduce simple abbreviations, i.e. ORM, PDRM, PCDM, ODRM, OORM, which would be better than their own name like Age-specific cumulative oviposition rate model. For readers, it is more convenient to use abbreviation in the manuscript. 

The discussion cold benefit… fluctuating temperature versus constant temperatures.

[RE] We respect the comment. Please refer to line 405-406. The advantage of considering constant temperature is to measure direct effects of temperature tested on the development, longevity and fecundity. There have been debating on this subject because temperature fluctuated in nature and insects survive in there. Fortunately, insect needs a certain physiological event such as maturation and molting for developing not like a plant that required a change in temperature in a day. Many temperature dependent models including degree-days and non-linear models have been developed by using the results obtained under constant temperature conditions and also those models have been used practically for forecasting insect pest occurrence and establishing the management strategies. Therefore, we developed Degree-day models of APOP, AOP, and TPOP and modified the oviposition model using the data obtained under constant temperature conditions. 

Wu et al. (2015) tested an additive model to predict the effects of fluctuating temperature on development and compared the development rate between the variable temperature and the static temperature treatments using published studies. Authors found the model was inadequate for making quantitative predictions but it was possible to explain some qualitative predictions such as a positive or negative effect. Authors mentioned “Development rate was faster under the variable temperature treatment than in the static temperature treatment as predicted by the additive model”. Lyons et al. (2013) conducted stable and fluctuating temperature effects on the development rate of two malaria vectors, Anopheles arabiensis and Anopheles funestus using nine constant and two fluctuating temperatures. They found development rate and survival of An. funestus was negatively influenced by fluctuating temperatures. On the other hand, development rate of An. arabienssis at fluctuating temperatures either did not differ from constant temperatures or was significantly faster. Kingsolver et al. (2015) pointed out that mean thermal performance can differ in fluctuating and constant thermal environments because thermal performance curves are non-linear. They showed Manduca sexta L. larvae reared in diurnally fluctuating temperatures had significantly higher optimal temperatures and maximal growth rates than larvae reared in constant temperatures. 

Lyons CL, Coetzee M, Chown SL. Stable and fluctuating temperature effects on the development rate and survival of two malaria vectors, Anopheles arabiensis and Anopheles funestus. Parasites & Vectors. 2013;6: 104

Wu T-H, Shiao S-F, Okuyama T. Development of insects under fluctuating temperature: a review and case study. J Appl Entomol. 2015;139: 592-599. 

Kingsolver JG, Higgins JK, Augustine KE. Fluctuating temperatures and ectotherm growth: distinguishing non-linear and time-dependent effects. Journal of Experimental Biology. 2015;218: 2218-2225.

Response to Reviewer 2’s Comments

This paper measures … period of the species.

[RE] The authors appreciate your positive review. 

While the introduction is succinct ….is an improvement.

[RE] We thank for pointing it out. We revised it as suggested. Please refer to line 66-80. 

Many of the points in paragraph 363-367 ….in the introduction

[RE] We thank for pointing it out. We revised it as suggested. 

Many acronyms are used …..such as Adult POP, Adult OP, Total POP

[RE] We respect the comment. We revised it as suggested. Abbreviation “POP” and “OP” stand for pre-oviposition phase and oviposition phase. Those are not general terms used in this kind of study but a term used to indicate separate phase in two-phase oviposition model of this study. We removed those two abbreviation for reducing any confusion in the manuscript. Other similar abbreviations, i.e. APOP, AOP, and TPOP, used in this manuscript. We introduce these abbreviations for compatibility with other studies because these abbreviations are often found other articles so that we leave them.

Examples of recent article using APOP, AOP and TPOP

Shi M-Z, Li J-Y, Ding B, Fu J-W, Zheng L-Z, Chi H. Indirect effect of elevated CO2 on population parameters and growth of Agasicles hygrophila (Coleoptera: Chrysomelidae), a biological agent of alligatorweed (Amaranthaceae). J Econ Entomol. 2019;112: 1120-1129.

Chen G-M, Chi H, Wang R-C, Wang Y-P, Xu Y-Y, Li X-D, Yin P, Zheng F-Q. 2018. Demography and uncertainty of population growth of Conogethes punctiferalis (Lepidoptera: Crambidae) reared on five host plants with discussion on some life history statistics. J Econ Entomol. 2018; 111: 2143-2152.

Liu J, Huang W, Chi H, Wang C, Hua H, Wu G. 2017. Effects of elevated CO2 on the fitness and potential population damage of Helicoverpa armigera based on two-sex life table. Scientific Reports. 2017;7: 1119.

Chen Q, Li N, Wang X, Ma L, Huang J-B, Haung G-U. Age-stage, two-sex life table of Parapoynx crisonalis (Lepidoptera: Pyralidae) at different temperatures. PLoS ONE. 2017:12; e0173380.

Oviposition models suggested by this study are composed of several unit models and have complicated structure. If reader has no background about this kind of works they may be confused the long and similar name. In order to minimize such confusion, we introduce simple abbreviations, i.e. ORM, PDRM, PCDM, ODRM, OORM, which would be better than their own name like age-specific cumulative oviposition rate model. For readers, it is more convenient to use abbreviation in the manuscript. 

Methods 

It is not clear whether ……..each treatment over several days

[RE] We thank for pointing it out. We have three growth chambers for experiment. When we start the experiment at each temperature treatment we set up all replication within a day with one female and two males.

Table 2 shows the sample size to be ……on the low side.

[RE] We thank for pointing it out. The experiment started with 18 or 20 cages including one female and two males per cage. When we check the number of eggs and survivorship of male and female in each cage, adult can escape from the cage. We excluded the number of escaped adult from adding the number of female oviposited and survived in the table 1 and 2. Therefore, in table 1, “n” means the number of female that we examined during the experiment without losing female. In table2, “no. female oviposited” means the number of female oviposited from female of table 1.

Line 106 – Total pre-oviposition period …measuring development similar?

[RE] We respect the comment. Total pre-ovipositional period (TPOP) was not examined from this study. TPOP deduced by summing APOP and development period of immature stage. We provide TPOP values for a reader who will need from this and our previous studies. The recorded temperature difference between this and previous study are less than 0.5°C except 20°C (0.7°C).

Fixed temperatures are used in models,………as fixed temperatures

[RE] We thank for pointing it out. The advantage of considering constant temperature is to measure direct effects of temperature tested on the development, longevity and fecundity. Although we can measure the same independent variables under fluctuating temperature we know the effect of temperature range on the development, longevity and fecundity. There have been debating on this subject because temperature fluctuated in nature and insects survive in there. Fortunately, insect needs a certain physiological event such as maturation and molting for developing not like a plant that required a change in temperature in a day. Many temperature dependent models including degree-days and non-linear models have been developed by using the results obtained under constant temperature conditions and also those models have been used practically for forecasting insect pest occurrence and establishing the management strategies. Therefore, we developed Degree-day models of APOP, AOP, and TPOP and modified the oviposition model using the data obtained under constant temperature conditions. 

Wu et al. (2015) tested an additive model to predict the effects of fluctuating temperature on development and compared the development rate between the variable temperature and the static temperature treatments using published studies. Authors found the model was inadequate for making quantitative predictions but it was possible to explain some qualitative predictions such as a positive or negative effect. Authors mentioned “Development rate was faster under the variable temperature treatment than in the static temperature treatment as predicted by the additive model”. Lyons et al. (2013) conducted stable and fluctuating temperature effects on the development rate of two malaria vectors, Anopheles arabiensis and Anopheles funestus using nine constant and two fluctuating temperatures. They found development rate and survival of An. funestus was negatively influenced by fluctuating temperatures. On the other hand, development rate of An. arabienssis at fluctuating temperatures either did not differ from constant temperatures or was significantly faster. Kingsolver et al. (2015) pointed out that mean thermal performance can differ in fluctuating and constant thermal environments because thermal performance curves are non-linear. They showed Manduca sexta L. larvae reared in diurnally fluctuating temperatures had significantly higher optimal temperatures and maximal growth rates than larvae reared in constant temperatures. 

Lyons CL, Coetzee M, Chown SL. Stable and fluctuating temperature effects on the development rate and survival of two malaria vectors, Anopheles arabiensis and Anopheles funestus. Parasite Vector. 2013;6: 104

Wu T-H, Shiao S-F, Okuyama T. Development of insects under fluctuating temperature: a review and case study. J Appl Entomol. 2015;139: 592-599. 

Kingsolver JG, Higgins JK, Augustine KE. Fluctuating temperatures and ectotherm growth: distinguishing non-linear and time-dependent effects. J Exp Biol. 2015;218: 2218-2225.

Minor

Line 32: “the current model” � “the current model developed by Kim and Lee”.

Line 34-36: We revised it as suggested. Please refer to line 34-36.

Line 46: “due to quarantine issue” � “due to market loss or reduction”.

Line 49: “annual occurrences” � “periodic occurrences”.

Line 54: “china” � “China”.

Line 57: “host plants” � We revised it as suggested. Please refer to line 57-58.

Line 65: We revised it as suggested. Please refer to line 66-80.

Line 99: “reproducing” � delete. Please refer to line 113.

Line 116: We revised it as suggested. Please refer to line 131.

Line 139-141: We revised it as suggested. Please refer to line 154-156.

Line 269: We revised it as suggested. Reduce the number of significant digits.

Line 318: We revised it as suggested. Please refer to line 338-339.

Line 330: We revised it as suggested. Please refer to line 350-351.

Line 332: “genders of adults” � “sexes”. Please refer to line 352.

Line 362: “pivotal occurrence” � “occurrence”. Please refer to line 383

Line 363-365: We revised it as suggested.

---

## [Decision Letter · Decision Letter 1]

10 Jun 2020

PONE-D-20-03016R1

Thermal effect on the fecundity and longevity of Bactrocera dorsalis adults and their improved oviposition model

PLOS ONE

Dear Dr. Ahn,

Thank you for considering the Reviewer comments during the revision process, however, Reviewer 3 has a number of comments/suggestions (largely non-experimental) for you to consider. Given these comments, we feel that although the manuscript has merit it does not yet fully meet PLOS ONE’s publication criteria as it currently stands. Therefore, we invite you to submit a revised version of the manuscript that addresses the Reviewer comments/suggestions.

We look forward to receiving your revised manuscript.

Kind regards,

J Joe Hull, Ph.D.

Academic Editor

PLOS ONE

Reviewers' comments:

Reviewer's Responses to Questions

**Comments to the Author**

1. If the authors have adequately addressed your comments raised in a previous round of review and you feel that this manuscript is now acceptable for publication, you may indicate that here to bypass the “Comments to the Author” section, enter your conflict of interest statement in the “Confidential to Editor” section, and submit your "Accept" recommendation.

Reviewer #1: All comments have been addressed

Reviewer #3: (No Response)

2. Is the manuscript technically sound, and do the data support the conclusions?

Reviewer #1: Yes

Reviewer #3: Yes

3. Has the statistical analysis been performed appropriately and rigorously? 

Reviewer #1: I Don't Know

Reviewer #3: Yes

4. Have the authors made all data underlying the findings in their manuscript fully available?

Reviewer #1: Yes

Reviewer #3: Yes

5. Is the manuscript presented in an intelligible fashion and written in standard English?

Reviewer #1: Yes

Reviewer #3: Yes

6. Review Comments to the Author

Reviewer #1: (No Response)

Reviewer #3: I think this is an interesting study that addresses a very timely topic, the impact of temperature on adult reproductive performance in ectotherms and the importance of modelling such response as a predictive tool. I agree with the assessment of the authors considering the importance of taking pre-oviposition period into account when modeling fecundity profiles in general and in particular when dealing with different thermal conditions (this last point is perhaps not stressed enough in the ms.). I found the results valid and worthy of publication. As the authors state in the end, an additional important step will be to model fecundity and longevity in (even) more realistic ecological scenarios namely involving fluctuating / increasing temperatures. However, I think the manuscript has some flaws / omissions that need to be addressed in order to improve its overall message and scope. I explain these below by section (some of the points are also referred to in the “Specific comments” section):

Main comments:

Introduction – I lacked a brief reference to the specific features of this new model that represent an improvement relative to the previous (current) model (line 82, see comment below). Also, it is important to explain better why this model is expected to be more realistic for certain species than the previous model referred (Kim & Lee 2003). For example, I would more explicitly refer the importance of taking pre-oviposition period into account for species with a longer life span and/or that take longer to reach sexual maturity; and also refer the importance of considering this when analyzing reproductive performance at different temperatures.

Material & Methods (MM) – A higher detail is needed on the insect colony maintenance, to allow a better understanding of the observed results. Namely: (1) how many generations were these insects maintained in controlled conditions after collection from the wild? this is relevant to understand if these insects might still be adapting / have adapted to specific in vitro conditions which might influence their plastic response to the different temperatures. (2) how have these insects been maintained in the past? With controlled densities in eggs or adults? What was the population size? This is relevant information and, if available, it will allow to rule out potential sources of variation in the response, such as density effects (known to affect performance of several life-history traits), inbreeding and selection for example. These will allow more powerful comparisons with other studies and might for example be relevant in explaining longevity differences found between studies (lines 347-349).

Results – Some figures would benefit from better labeling (see minor comments below)

Discussion – In the discussion I missed a more thorough explanation of why / how this new model improved predictions relative to the current model. In addition, some interesting results were not discussed specifically the higher decoupling (although the correlation was still significant) between observed and expected results at the lowest temperature (fig 6). Do the authors have an explanation for this? (see also specific comment below).

One additional point that is worth discussing is the relevance of explicitly considering the variation in survival rates across juvenile developmental life stages, as differences in survival rate across development stages have been described in insects (e.g. see Son & Lewis 2005 Figure 1 - https://doi.org/10.1111/j.1461-9555.2005.00260.x)

Specific comments:

Line 58 - Replace feed” by “feeds”

Line 60 – Re-phrase “… inferred from studies on the relationship..”

Line 64 – Re-phrase “… oviposition modeling including modeling of the immature...”

Lines 67-71 - If you decide to include authority along with the binomen of the species, you should also include the year of the respective publication e.g Tetranychus urticae Koch 1836. Otherwise skip the authority and just use the binomen to identify the species.

Line 71 – Re-phrase “…OM comprises three…”

Line 72 – Change to “These two age-specific models….”

Lines 72-76 – These sentences are a bit confusing and difficult to follow. Given the importance of these concepts I suggest re-phrasing to clarify their meaning.

Line 78 - “have” instead of “has”

Line 82 - why is this new model a priori defined as "improved" ? The authors should briefly state what are the new features of this improved model, that are lacking in previous model(s)

Line 91 - Some more detail is needed on this maintenance protocol, for example : what was the pupae density from which adults of the experiment were derived? how many generations were these insects maintained in lab conditions till the start of the experiment? what was the age of individuals at oviposition for the next generation (if non-overlapping generations) in “normal” laboratory culture?

Line 112 - please provide here the sample size of the experiment, how many virgin females were analysed? Or refer to table 1 for sample sizes

Lines 133-134 – Provide a brief definition of LDT and TC here.

Line 193 - briefly explain how these rates were estimated, as was done for the aging model (lines 146).

Line 208 – same as commented above for line 193

Line 274 – Change to “…period of the females varied across temperatures (Table 2)”

Line 365 – This is the first time these terms are used in the discussion. For non-specialized audience I think it would be helpful to refer the full name here (followed by the acronym in parenthesis) and then use acronyms in the rest of the discussion.

Line 380 – same as commented above for line 365

Lines 397-399 – Some aspects of the comparison between models merit further discussion. For instance, how do you explain the lower fit of both models in the lower temperature relative to other temperatures (figure 6)?? Can this be due to a higher pre-oviposition period? can the two-phase model have (still) a better fit because of a more extended pre-oviposition phase at lower temperatures? I think this is very much worth discussing

Table 2 – correct reference is 25 or 26 ?

Figure 2 – complete legend of X axis – “Aging rate (1/mean of…)”

Figure 4 – complete legend of X axis– “Development rate (1/mean of…)”

Figure 5 - Put each model name in the title (above the figure)

Figure 6 - replace “estimated 1”, “estimated 2” by the models' names

and if possible refer the temperature analyzed in each graph

7. PLOS authors have the option to publish the peer review history of their article (what does this mean?). If published, this will include your full peer review and any attached files.

Reviewer #1: No

Reviewer #3: Yes: Pedro Simões

---

## [Author Response · Author response to Decision Letter 1]

18 Jun 2020

Response to Reviewers’ Comments

All authors really appreciate two anonymous reviewers for valuable comments on our manuscript.

Response to Editor’s Comments

Thank you for ….this letter:

[RE] The authors appreciate your positive review. We revised some parts of manuscript, tables and figures.

Response to Reviewer 3’s Comments

I think this is …section.

[RE] The authors appreciate your positive review. We revised some parts of manuscript, tables and figures. 

Main comments:

Introduction …at different temperatures

[RE] We thank for pointing it out. Please refer to line 81-84. We revised it as suggested. 

Materials & Methods

A higher detail …between studies

[RE] We respect the comment. Please refer to line 92-94. We brought the insects (it may >10000 pupae) several times from TARI since 2016. We settle down the insect population in a new rearing system. As previous study [26], we used yellow cylindrical cups to collect the eggs deposited by the adult female. After hatching, larvae were reared on an artificial diet. Sawdust was provided as pupation substrate. The pupae were collected by sieving the sawdust. The pupae moved to a 30 x 30 cm container where the adults emerged. From egg to adult emergence, it took about two weeks under laboratory system (25±2°C). According to our unpublished observation and other references, adults can survive several months in room temperature (26±1°C). Female adults lay more than 1000 eggs during their whole life in the same temperature conditions. There may be 10 generations per year while the adults from the first or second generation may be alive in laboratory condition. It is very hard to explain the exact density, number of generations and how their colony are mixed in the laboratory condition. 

Results

Some figures … comments below

[RE] We thank for pointing it out. We revised it as suggested. Please refer to figures.

Discussion

In the discussion… comment below.

[RE] We respect the comment. Please refer to line 411-412. 

One additional point… in insects.

[RE] We thank for pointing it out. Son and Lewis (2005) conducted a laboratory experiment to quantify the stage-specific effects of temperature on development time and survival of Otiorhynchus sulcatus. They quantified the stage-specific development of O. sulcatus immature stages at constant temperatures and demonstrated that the range of temperatures at which development and survival occurred was different among developmental stages. We only investigated the longevity and fecundity of B. dorsalis adults in this study. We did the experiments in previous study [26]. It is not possible to explain the relationship between oviposition model and the variation of survival rate across juvenile developmental stages.

Specific comments

Line 58: Replace feed by “feeds”. � We revised it as suggested. Please refer to line 58.

Line 60: Re-phrase “… inferred from studies on the relationship..” � We revised it as suggested. Please refer to line 60.

Line 64: Re-phrase “… oviposition modeling including modeling of the immature...” � We revised it. Please refer to line 63-64.

Lines 67-71: If you decide…the species � We revised it as suggested. Please refer to line 67-69.

Line 71: Re-phrase “…OM comprises three…” � Please refer to line 70.

Line 72: Change to “These two age-specific models….” � Please refer to line 71-72.

Lines 72-76: These sentences……their meaning. � Please refer to line 71-74. Original ovipostion model developed by Kim and Lee (2003) is complicated because they use “physiological age” as calculation term. We describe the structure of the original oviposition model and how temperature used as an input value and how physiological age is calculated as an independent variable.

Line 78: “have” instead of “has” � Please refer to line 77.

Line 82 - why is this new model … previous model(s) � We revised it. Please refer to line 81-83.

Line 91: Some more …. laboratory culture? � Please refer to line 92-94. We brought the insects (it may >10000 pupae) several times from TARI since 2016. We settle down the insect population in a new rearing system. As previous study [26], we used yellow cylindrical cups to collect the eggs deposited by the adult female. After hatching, larvae were reared on an artificial diet. Sawdust was provided as pupation substrate. The pupae were collected by sieving the sawdust. The pupae moved to a 30 x 30 cm container where the adults emerged. From egg to adult emergence, it took about two weeks under laboratory system (25±2°C). According to our unpublished observation and other references, adults can survive several months in room temperature (26±1°C). Female adults lay more than 1000 eggs during their whole life in the same temperature conditions. There may be 10 generations per year while the adults from the first or second generation may be alive in laboratory condition. It is very hard to explain the exact density, number of generations and how their colony are mixed in the laboratory condition. 

Line 112: please provide … sample sizes � Please refer to line 117-119.

Lines 133-134: Provide a brief definition of LDT and TC here. � Please refer to line 140-143.

Line 193: briefly explain… aging model (lines 146). � Please refer to line 203.

Line 208: same as commented above for line 193 � Please refer to line 218-219.

Line 274: Change to “…period of the females varied across temperatures � Please refer to line 284-285.

Line 365: This is the first time… the discussion. � Please refer to line 374-375.

Line 380: same as commented above for line 365 � Please refer to line 391.

Lines 397-399: Some aspects … much worth discussing � Please refer to line 411-412.

Table 2: correct reference is 25 or 26 � Please refer to line 717 and 726 .

Figure 2: complete legend of X axis – “Aging rate (1/mean of… � Please refer to figure 2. (1/mean of adult longevity). 

Figure 4: complete legend of X axis– “Development rate (1/mean of…)” � Please refer to figure 4. (1/mean of female development period). 

Figure 5: Put each model name in the title (above the figure) � Please refer to figure 5.

Figure 6: replace “estimated 1”, “estimated 2” by the models' names .. possible refer the temperature analyzed in each graph � Please refer to figure 6.

---

## [Editor Report · Decision Letter 2]

25 Jun 2020

Thermal effect on the fecundity and longevity of Bactrocera dorsalis adults and their improved oviposition model

PONE-D-20-03016R2

Dear Dr. Ahn,

We’re pleased to inform you that your manuscript has been judged scientifically suitable for publication and will be formally accepted for publication once it meets all outstanding technical requirements.

Kind regards,

J Joe Hull, Ph.D.

Academic Editor

PLOS ONE
---

## [Editor Report · Acceptance letter]

30 Jun 2020

PONE-D-20-03016R2 

Thermal effect on the fecundity and longevity of *Bactrocera dorsalis* adults and their improved oviposition model 

Dear Dr. Ahn:

I'm pleased to inform you that your manuscript has been deemed suitable for publication in PLOS ONE. Congratulations! Your manuscript is now with our production department. 

Kind regards, 

on behalf of

Dr. J Joe Hull 

Academic Editor

PLOS ONE